# ProMeCD: Unifying Long-Tailed and Noisy Label Learning via White-Box Control

**Yu Zhu** [1]

## Abstract

Real-world data is rarely clean; it is plagued by severe class imbalance (long-tailed distributions) and label corruption. Current solutions lean heavily on "black-box" meta-learning to re-weight samples. However, this paradigm introduces a fatal circular dependency: it relies on pristine, balanced validation sets to guide the optimization, which are essentially non-existent in the wild. We propose ProMeCD, a self-referential framework that breaks this dependency by recasting optimization as an autonomous control problem. Instead of training an opaque neural meta-learner, we employ a transparent proportional-integral controller. The system monitors "cognitive entropy" that is a metric derived from von Mises-Fisher gradient statistics to assess learning uncertainty. To resolve the scalar ambiguity between tail and noisy samples, ProMeCD employs a decoupled control strategy: it boosts tail classes via integral accumulation of magnitude deficits when directional consistency is high, while suppressing noise via proportional feedback when consistency collapses. Theoretically, we prove that this mechanism guarantees convergence and formally prevents the minority initial drop, ensuring monotonic improvement for rare classes. Crucially, ProMeCD is fully white-box and validation-free. Experiments on CIFAR-LT, iNaturalist, CIFAR-N, mini Web-Vision, and Clothing1M confirm that ProMeCD is not merely efficient; it outperforms the recent meta-learner FMW-Net by over 10% in severe imbalance settings, proving that explicit control theory offers a superior path to handling imperfect data.

[1]School of Science, Tianjin University of Commerce, Tianjin, China. Correspondence to: Yu Zhu <yuzhu@tjcu.edu.cn>.

*Proceedings of the $43^{rd}$ International Conference on Machine Learning*, Seoul, South Korea. PMLR 306, 2026. Copyright 2026 by the author(s).

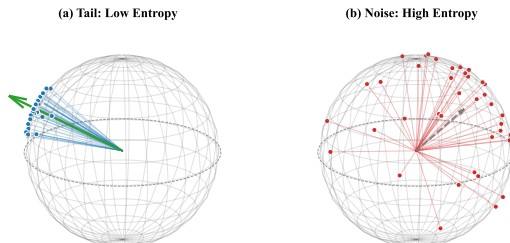

**(a) Tail: Low Entropy**  **(b) Noise: High Entropy**

*Figure 1.* Gradient cognitive dynamics. (a) Tail Class: Gradients cluster tightly (consistent), indicating low cognitive entropy. (b) Noisy Sample: Gradients scatter chaotically, indicating high cognitive entropy.

## 1. Introduction

The remarkable success of deep learning is largely predicated on the availability of large-scale, meticulously annotated datasets (Deng et al., 2009; Krizhevsky et al., 2012). However, this idealized scenario rarely holds in the wild (Sun et al., 2017). Real-world data is inherently imperfect, frequently suffering from a spectrum of defects, most notably long-tailed class distributions (Cui et al., 2019) and label noise (Johnson & Khoshgoftaar, 2019; Frénay & Verleysen, 2013). In long-tailed settings, a few "head" classes are over-represented while the majority of "tail" classes have scant samples (Liu et al., 2019; Tang et al., 2020). This severe class imbalance causes models trained with standard objectives, such as cross-entropy loss, to develop a strong bias towards the head classes (Byrd & Lipton, 2019; Kang et al., 2019), leading to abysmal performance on the tail (Zhang et al., 2023b). In noisy label scenarios, models are prone to overfitting to incorrect labels due to their high capacity and propensity to memorize noisy data (Zhang et al., 2021a; Arpit et al., 2017), resulting in poor generalization ability (Song et al., 2022; Li et al., 2019).

To tackle these challenges, research has evolved along several lines. The "static compensation" paradigm employs one-off correction strategies, such as data re-sampling (Chawla et al., 2002; Buda et al., 2018; He & Garcia, 2009) or loss function re-weighting (Lin et al., 2017; Menon et al., 2020; Cao et al., 2019). While beneficial, these methods cannot adapt to the ever-changing dynamics of the training process. In pursuit of greater adaptivity, the "black-box" meta-learning paradigm has emerged. Methods like L2R

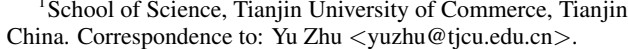

(Ren et al., 2018) use a neural network (the meta-learner) to learn sample weights, guided by performance on a clean validation set. While more flexible, this approach has three critical drawbacks: (1) a strong dependency on a clean, unbiased validation set, which is often unavailable in practice (Li et al., 2020); (2) high computational overhead from complex bi-level optimization (Shu et al., 2019); and (3) poor interpretability of the black-box decision process (Rudin, 2019; Zhang et al., 2021b).

Existing re-weighting methods often struggle with a fundamental dilemma: scalar ambiguity. Both rare tail samples and corrupted noisy samples produce similarly high loss values, making them indistinguishable to a standard optimizer. Indiscriminately boosting high-loss samples causes overfitting to noise, while suppressing them leads to underfitting on the tail. To resolve this impasse, we turn to the "dynamic analysis" paradigm (Francazi et al., 2023), which suggests that the essence of performance degradation lies not in the scalar loss, but in pathological gradient dynamics.

Figure 1 visualizes this critical distinction. While both types of samples may exhibit weak gradient magnitudes, their structural consistency differs fundamentally. Tail samples (Figure. 1(a)) exhibit consistent directionality towards the optimization target (low entropy), whereas noisy samples (Figure. 1(b)) generate conflicting, chaotic signals (high entropy). This insight inspires a new direction: an ideal solution should be able to perceive these vector-level dynamics in real-time and control them autonomously.

This is the core departure point of our work. We ask the central scientific question: Can we design an optimization framework that possesses the adaptivity of meta-learning, but without requiring a clean validation set, while remaining efficient, interpretable, and theoretically sound?

To answer this, we propose probabilistic meta-learning of cognitive dynamics (ProMeCD), a novel "white-box" meta-learning framework. ProMeCD's key innovation is the replacement of the black-box neural meta-learner with a classic, efficient, and fully interpretable proportional-integral (PI) controller from control theory as its meta-decision algorithm.

1. **Perception:** ProMeCD probabilistically models per-class gradients using von Mises-Fisher (vMF) distributions and quantifies the learning state via our proposed "cognitive entropy".

2. **Decision:** The decoupled PI controller analytically computes gradient modulation coefficients by separately sensing learning quantity (magnitude $m$) and quality (consistency $\kappa$), allowing for opposing control actions on indistinguishable high-loss samples.

3. **Action:** The model uses these coefficients to weight

gradients, completing a cycle of autonomous correction.

As a fully self-referential system, ProMeCD's feedback loop is driven by the internal properties of the training process (the stability of gradient distributions), thus obviating the need for external validation sets. We provide a solid theoretical foundation for this framework, proving its convergence and its ability to prevent pathological behaviors like the minority initial drop (MID). Comprehensive experiments, including ablations against ProMeCD-P/PID variants and systematic comparisons with black-box meta-learners like MW-Net (Shu et al., 2019) and FMW-Net (Zhou et al., 2025), demonstrate the superior trade-off of our "white-box" meta-learning approach in terms of performance, efficiency, generality, and interpretability.

## 2. Related Work

### 2.1. Learning from Defective Data

Traditional approaches handle long-tails via re-balancing (Chawla et al., 2002; Cui et al., 2019; Menon et al., 2020) or label noise via robust selection and losses (Zhang & Sabuncu, 2018; Han et al., 2018; Wei et al., 2020), mostly as static compensation. Recent joint solvers such as D-SINK (Hong et al., 2026), RCAL (Zhang et al., 2023a), and TABASCO (Lu et al., 2023) address dual defects within the regime of Long-Tailed Noisy Label Learning (LTNLL). However, these specialized solutions frequently depend on complex multi-stage heuristics or task-specific pipelines, which can limit their generalizability across diverse data manifolds and architectures.

### 2.2. Control-Theoretic Optimization

Incorporating control theory into deep learning is promising but distinct from ProMeCD. Existing methods design novel optimizers (e.g., PID-Optimizer (Yu et al., 2024) for parameter-level updates) or schedulers (e.g., PID-GAN (Chen et al., 2024) for global learning rates). In contrast, ProMeCD operates at the cognitive level. Instead of using raw gradients or model outputs as signals, it utilizes cognitive entropy to regulate per-class gradient contributions, acting as a general-purpose dynamics stabilizer.

### 2.3. Meta-Learning for Reweighting

"Black-box" meta-learning (e.g., MW-Net (Shu et al., 2019)) relies on auxiliary networks trained via bi-level optimization on clean validation sets. While flexible, these methods suffer from high computational overhead, poor interpretability, and dependency on external data. ProMeCD introduces a "white-box" paradigm: it replaces the neural meta-learner with an interpretable analytical PI controller and relies on

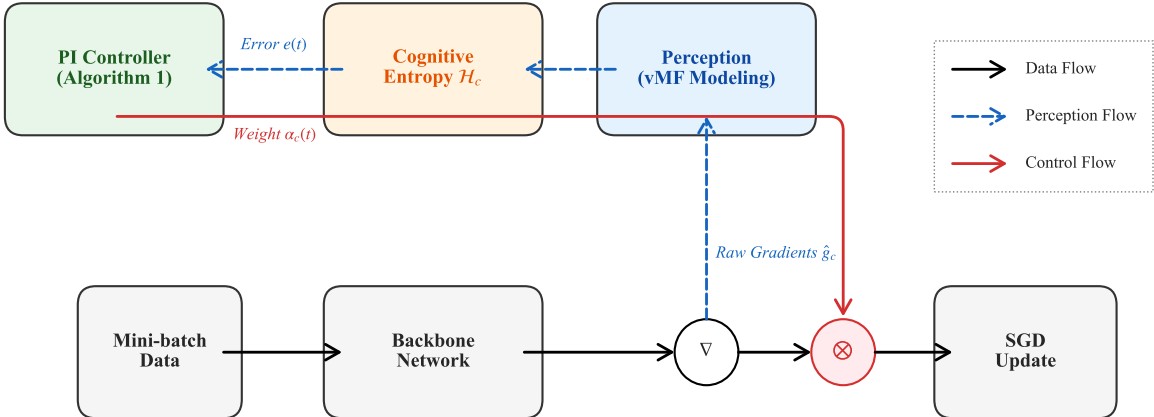

*Figure 2.* The ProMeCD framework as a closed-loop system. The standard optimization path (bottom, black) is intercepted by a meta-control loop (top). The system perceives gradient uncertainty via cognitive entropy (blue dashed flow), decides on a correction strategy using the PI controller, and acts by modulating gradients (red solid flow) before the SGD update. This design enables autonomous adaptation without external validation sets.

endogenous self-referential feedback, thereby eliminating the need for clean validation sets.

## 3. The ProMeCD Framework

As illustrated in Figure 2, ProMeCD reimagines the optimization landscape as a feedback control loop, fundamentally differing from traditional open-loop training. The framework operates by intercepting the raw gradients generated from the standard forward pass to assess the "cognitive state" of the learning process in real-time.

Instead of applying gradients directly, the system first routes them through a perception module, which models their directional statistics via vMF distributions to quantify uncertainty in the form of cognitive entropy. This entropy signal serves as the input error for the PI controller, which acts as the system's decision core. By calculating the deviation from an ideal low-entropy state, the controller analytically derives a modulation coefficient $\alpha_t$. Finally, the loop is closed via an action mechanism (red solid flow), which injects this coefficient back into the optimization stream at the multiplication node ($\otimes$). This allows the model to dynamically boost under-represented patterns or suppress noisy signals before the parameters are updated, ensuring that the optimization trajectory is actively guided by the stability of the learning dynamics itself.

### 3.1. Perception Module: Probabilistic Modeling of Gradient Dynamics

The foundation of ProMeCD is the real-time perception of the model's cognitive state. To capture the full cognitive state, we define the per-class gradient $\hat{\mathbf{g}}_c \in \mathbb{R}^d$ as the concatenation of flattened gradients from the last residual block and the classifier layer, where $d = \sum d_{\text{layer}}$ is the joint di-

mensionality. This allows ProMeCD to perceive uncertainty across both feature representation and class-specific logic. For large-scale or high-cardinality tasks such as iNaturalist 2018 and Clothing1M, we adopt a classifier-only proxy to circumvent hardware memory constraints while maintaining sufficient cognitive perception. We then model $\hat{\mathbf{g}}_c$ as a random variable from SGD.

#### 3.1.1. THE VON MISES-FISHER (VMF) PRIOR FOR GRADIENT DIRECTIONS

To formally model the stochastic nature of gradient directions, we hypothesize that the normalized gradient direction, $\mathbf{u}_c = \hat{\mathbf{g}}_c/\|\hat{\mathbf{g}}_c\|$, which is a unit vector residing on the hypersphere $S^{d-1}$ (where $d$ is the joint dimensionality defined in Section. 3.1), follows a vMF distribution. The vMF distribution is a probability distribution on the $(d-1)$-sphere that is a natural analogue to the multivariate normal distribution. Its probability density function is given by:

$$p(\mathbf{u}_c|\boldsymbol{\mu}_c, \kappa_c) = C_d(\kappa_c)\exp(\kappa_c\boldsymbol{\mu}_c^T\mathbf{u}_c), \qquad (1)$$

where $C_d(\kappa_c) = \kappa_c^{d/2-1}/((2\pi)^{d/2}I_{d/2-1}(\kappa_c))$ is the normalization constant, with $I_\nu$ being the modified Bessel function of the first kind of order $\nu$. The parameters $\boldsymbol{\mu}_c$ and $\kappa_c$ are referred to as the mean direction and concentration parameter, respectively. The mean direction $\boldsymbol{\mu}_c$ (where $\|\boldsymbol{\mu}_c\|_2 = 1$) indicates the central tendency or the most likely direction of the gradient vectors for class $c$. The concentration parameter $\kappa_c$ ($\kappa_c \geq 0$) determines how tightly the gradient directions are clustered around this mean direction. A higher concentration around the mean direction $\boldsymbol{\mu}_c$ is observed with greater $\kappa_c$, signifying a stable and consistent learning signal with low directional noise. Conversely, as $\kappa_c$ approaches zero, the distribution becomes uniform on the sphere, representing a state of maximum directional un-

certainty where the gradient signal is essentially random noise.

### 3.1.2. ONLINE ESTIMATION OF COGNITIVE DYNAMIC METRICS

This probabilistic model gives us a tuple of cognitive dynamic metrics $S_c(t) = (\boldsymbol{\mu}_c(t), \kappa_c(t), m_c(t))$. To estimate these online, we maintain an exponential moving average (EMA) of the raw per-class gradients:

$$\bar{\mathbf{g}}_c(t) = (1 - \gamma)\bar{\mathbf{g}}_c(t-1) + \gamma\hat{\mathbf{g}}_c(t), \tag{2}$$

where $\hat{\mathbf{g}}_c(t)$ denotes the observed mean gradient of class $c$ computed from the current mini-batch, $\bar{\mathbf{g}}_c(t)$ represents the smoothed gradient estimation accumulating historical information, and $\gamma \in (0, 1)$ is the smoothing coefficient (or momentum) controlling the update rate. From the smoothed gradient estimation $\bar{\mathbf{g}}_c(t)$, and following the Maximum Likelihood Estimation (MLE) for a $d$-dimensional vMF distribution, we define the mean direction estimator $\mu_c(t)$ and the learning quantity $m_c(t)$ as:

$$\mu_c(t) = \frac{\bar{\mathbf{g}}_c(t)}{\|\bar{\mathbf{g}}_c(t)\|_2}, \quad m_c(t) = \|\bar{\mathbf{g}}_c(t)\|_2 \tag{3}$$

where $\bar{\mathbf{g}}_c(t)$ acts as the recursive resultant vector accumulating gradient information. Physically, $m_c(t)$ represents the effective signal strength, serving as the primary feedback for the PI controller. The concentration parameter $\kappa_c(t)$ is efficiently approximated from the resultant vector length based on the ratio of modified Bessel functions (Sra, 2012). To maintain the self-containment of our white-box framework, we provide the explicit approximation formula for $\kappa$ and its numerical stability considerations in Appendix A.

## 3.2. Objective Module: Information-Theoretic Cognitive Entropy

Having established a probabilistic model for gradient dynamics, we require a principled, scalar objective function for our controller. Simply combining the metrics $m_c$ and $\kappa_c$ via a weighted sum is heuristic and lack theoretical grounding. Instead, we propose a unified measure, cognitive entropy, denoted $\mathcal{H}_C(c)$, which quantifies the total uncertainty of the learning state for a class $c$ from a fundamental information-theoretic perspective.

A pathological learning state is fundamentally a state of high uncertainty. This uncertainty arises from two orthogonal sources: (1) uncertainty about the strength of the learning signal, and (2) uncertainty about the direction of the learning signal. Cognitive entropy is defined as the sum of these two uncertainties.

### 3.2.1. QUANTIFYING DIRECTIONAL UNCERTAINTY VIA DIFFERENTIAL ENTROPY

The uncertainty in the gradient direction is naturally captured by the differential entropy of its underlying probability distribution. For a random unit vector $\mathbf{u}$ following a vMF distribution $p(\mathbf{u}|\boldsymbol{\mu}, \kappa)$, its differential entropy is defined as $H_{\text{vMF}} := -\mathbb{E}_{\mathbf{u}\sim p}[\log p(\mathbf{u}|\boldsymbol{\mu}, \kappa)]$. We can derive its analytical form as follows.

**Proposition 3.1** (Differential Entropy of vMF). *The differential entropy of a $d$-dimensional vMF distribution with concentration $\kappa$ is:*

$$H_{vMF}(\kappa) = -\log C_d(\kappa) - \kappa A_d(\kappa), \tag{4}$$

*where $C_d(\kappa)$ is the normalization constant and $A_d(\kappa) = I_{d/2}(\kappa)/I_{d/2-1}(\kappa)$.*

*Proof.* By definition, differential entropy is $H = -\mathbb{E}_{\mathbf{u}}[\log p(\mathbf{u}|\boldsymbol{\mu}, \kappa)]$. Substituting the log-likelihood of the vMF distribution:

$$H = -\mathbb{E}\left[\log C_d(\kappa) + \kappa\boldsymbol{\mu}^\top\mathbf{u}\right] = -\log C_d(\kappa) - \kappa\boldsymbol{\mu}^\top\mathbb{E}[\mathbf{u}]. \tag{5}$$

A fundamental property of the vMF distribution is that its first moment is aligned with the mean direction: $\mathbb{E}[\mathbf{u}] = A_d(\kappa)\boldsymbol{\mu}$. Since $\|\boldsymbol{\mu}\| = 1$, the inner product simplifies to $\boldsymbol{\mu}^\top\mathbb{E}[\mathbf{u}] = A_d(\kappa)\|\boldsymbol{\mu}\|^2 = A_d(\kappa)$. Substituting this back yields the proposition. $\square$

The term $A_d(\kappa)$ has a clear physical interpretation: it is the mean resultant length, representing the average alignment of the random vectors with the mean direction. It is a monotonic function with $A_d(0) = 0$ (for a uniform distribution) and $\lim_{\kappa\to\infty} A_d(\kappa) = 1$ (for a perfectly concentrated distribution). Thus, a high concentration $\kappa_c$ leads to a high $A_d(\kappa_c)$, a low differential entropy $H_{\text{vMF}}$, and therefore low directional uncertainty.

### 3.2.2. QUANTIFYING MAGNITUDE UNCERTAINTY VIA LOGARITHM

Directional uncertainty only captures the consistency of the signal, not its strength. A learning process can be directionally consistent (high $\kappa_c$) but still be ineffective if the gradient magnitude $m_c$ is vanishing. This represents an uncertainty about whether the model is making any meaningful progress. In information theory, the standard way to represent the uncertainty of a positive scalar value like magnitude is through its negative logarithm. This gives us the magnitude uncertainty term: $-\log m_c$. This term is high when the signal is weak ($m_c \to 0$) and low when the signal is strong.

### 3.2.3. DEFINITION OF COGNITIVE ENTROPY

By combining these two principled measures of uncertainty, we arrive at the definition of cognitive entropy.

**Definition 3.2** (Cognitive Entropy). The cognitive entropy $\mathcal{H}_C(c)$ for a class $c$ is defined as the normalized sum of its magnitude and directional uncertainties:

$$\mathcal{H}_C(c) := \frac{1}{d}\left[\underbrace{-\log(m_c + \epsilon)}_{\text{Magnitude Uncertainty}} + \underbrace{H_{\text{vMF}}(\kappa_c)}_{\text{Directional Uncertainty}}\right] \quad (6)$$

where $\epsilon = 10^{-6}$ is a stability constant.

Normalizing by $1/d$ ensures architectural and scale invariance. As the total differential entropy scales linearly with the dimensionality $d$, this normalization ensures that $\mathcal{H}_C(c)$ remains on a consistent scale regardless of the specific layers selected for perception. Consequently, it allows ProMeCD to maintain consistent control gains $(\beta_p, \beta_i)$ whether $d$ includes the full backbone or only a "classifier-only" proxy.

Cognitive entropy captures the two primary failure modes of a learning signal: being too weak (low $m_c$) and being too noisy/inconsistent (low $\kappa_c$). A learning state is only considered "healthy" (low entropy) when both magnitude and concentration are high. Consequently, minimizing cognitive entropy naturally encourages a state of high certainty, where learning signals are both strong (high magnitude) and stable (high concentration).

**Resolving Scalar Ambiguity.** A fundamental limitation of standard optimization is the scalar ambiguity of the loss function. Standard loss assigns high values to both tail and noisy samples, rendering them indistinguishable. In contrast, ProMeCD resolves this via cognitive entropy (Equation (6)). By leveraging gradient vector statistics, it effectively disentangles these cases: tail samples manifest as low entropy (warranting an integral boost), whereas noisy samples remain high entropy (requiring proportional suppression).

### 3.3. Control Module: Decoupled PI Controller

To resolve the scalar ambiguity where tail and noisy samples both exhibit high entropy, we decouple the PI controller into two specialized feedback loops. This design treats gradient magnitude $m_c(t)$ as the quantity of learning and concentration $\kappa_c(t)$ as the quality of learning. The modulation coefficient $\alpha_c(t)$ for class $c$ at step $t$ is computed as:

$$\alpha_c(t) = \max\left(0, 1 + \beta_i I_c(t) - \beta_p P_c(t)\right), \quad (7)$$

where $\beta_i, \beta_p > 0$ are gains. The terms are defined recursively to facilitate online optimization:

**1. Integral Loop ($I_c(t)$):** This loop corrects the "learning lag" of rare classes. To prevent sensitivity to gradient outliers, we define the target magnitude $\bar{m}(t)$ as the top-5% quantile of all class-wise magnitudes at step $t$:

$$\bar{m}(t) = \text{Quantile}_{0.95}(\{m_j(t)\}_{j=1}^C). \quad (8)$$

The integral term $I_c(t)$ is updated as:

$$I_c(t) = \text{clip}\left(I_c(t-1) + \Delta I_c(t), 0, I_{\max}\right), \quad (9)$$

where $\Delta I_c(t) = \max(0, \bar{m} - m_c) \cdot \mathbb{I}(\kappa_c > \tau_\kappa)$. The constant $I_{max}$ acts as an anti-windup mechanism to prevent the controller from overshooting after prolonged learning lags. $\tau_\kappa = \mathbb{E}_j[\kappa_j(t)]$ is the dynamic consistency threshold, defined as the moving average of concentration across all classes to ensure validation-free adaptation.

**2. Proportional Loop ($P_c(t)$):** This loop provides an immediate penalty for directional chaos. $P_c(t)$ measures the consistency collapse below $\tau_\kappa$:

$$P_c(t) = \max(0, \tau_\kappa - \kappa_c(t-1)). \quad (10)$$

Specifically, the dynamic threshold $\tau_\kappa$ defines a dataset noise floor relative to the current manifold maturity, distinguishing valid feature diversity from chaotic noise. When $\kappa_c(t-1) < \tau_\kappa$, the penalty $P_c(t)$ increases, allowing the negative feedback in Equation (7) to suppress the gradient weight, potentially driving $\alpha_c(t) \to 0$ for corrupted data.

## 4. Theoretical Analysis

In this section, we provide a rigorous analysis of the convergence properties of ProMeCD and its connection to generalization. We first establish the conditions under which the proposed control mechanism guarantees monotonic loss reduction, and then extend the analysis to the stochastic setting.

### 4.1. Deterministic Analysis: Monotonic Convergence and MID Prevention

We first analyze the behavior of ProMeCD in the deterministic (full-batch) setting to demonstrate how it formally prevents the Minority Initial Drop (MID).

**Lemma 4.1** (Loss Change under Modulated Gradient Descent). *Assume that the per-class loss function $L_c(\mathbf{x})$ is $L$-smooth with respect to the model parameters $\mathbf{x}$. Let $\mathbf{g}_c = \nabla L_c(\mathbf{x}_t)$ denote the true gradient for class $c$ at step $t$. Let $w_j = |\mathcal{B}_{t,j}|/|\mathcal{B}_t|$ be the normalized weight of class $j$ in the mini-batch. Under the ProMeCD update rule $\mathbf{x}_{t+1} = \mathbf{x}_t - \eta \sum_{j=1}^C \alpha_j(t) w_j \mathbf{g}_j$, the change in loss for a specific class $c$, denoted as $\Delta L_c = L_c(\mathbf{x}_{t+1}) - L_c(\mathbf{x}_t)$,*

**Algorithm 1** The ProMeCD Framework

1: **Input:** Training data $\mathcal{D}$, learning rate $\eta$, gains $\beta_p, \beta_i$, threshold $\tau_\kappa$.
2: **Initialize:** Model parameters $\mathbf{x}_0$, EMA stats $\bar{\mathbf{g}}_c = \mathbf{0}$, integral error $I_c = 0$.
3: **for** $t = 0, 1, 2, ...$ **do**
4:     Sample mini-batch $\mathcal{B}_t$.
5:     // — 1. Perception Module —
6:     **for** each class $c \in \mathcal{B}_t$ **do**
7:         Update EMA $\bar{\mathbf{g}}_c(t)$ and estimate $m_c(t), \kappa_c(t)$.
8:     **end for**
9:     // — 2. Control Module (Delayed Action) —
10:    Set target magnitude
      $\bar{m}(t - 1) = \text{Quantile}_{0.95}(\{m_j(t-1)\}_{j=1}^C)$.
11:    **for** each class $c = 1, ..., C$ **do**
12:       **if** $\kappa_c(t - 1) > \tau_\kappa$ **then**
13:          $I_c(t) \leftarrow \text{clip}\left(I_c(t-1) + \Delta I_c(t), 0, I_{\max}\right)$.
14:          $P_c(t) \leftarrow 0$.
15:       **else**
16:          $I_c(t) \leftarrow I_c(t-1)$.
17:          $P_c(t) \leftarrow \tau_\kappa - \kappa_c(t-1)$.
18:       **end if**
19:       $\alpha_c(t) = \max(0, 1 + \beta_i I_c(t) - \beta_p P_c(t))$.
20:    **end for**
21:    // — 3. Action Module —
22:    $\mathbf{g}_{\text{ProMeCD}} = \sum_{c \in B_t} \alpha_c(t) w_{t,c} \hat{\mathbf{g}}_c(t)$.
23:    $\mathbf{x}_{t+1} = \mathbf{x}_t - \eta \mathbf{g}_{\text{ProMeCD}}$.
24: **end for**

---

*satisfies the following inequality:*

$$\Delta L_c \leq -\eta \left\langle \mathbf{g}_c, \sum_{j=1}^C \alpha_j(t) w_j \mathbf{g}_j \right\rangle + \frac{L\eta^2}{2} \left\| \sum_{j=1}^C \alpha_j(t) w_j \mathbf{g}_j \right\|_2^2 . \quad (11)$$

*Proof.* See Appendix B.1.

**Theorem 4.2** (Monotonic Convergence Guarantee). *Let $w_j = |\mathcal{B}_{t,j}|/|\mathcal{B}_t|$ be the normalized weight of class $j$ in the mini-batch. If the modulation coefficients $\alpha_c(t) > 0$ generated by the PI controller satisfy the condition that the effective gradient ratio $C_{t,eff}^{(c)}$ is strictly less than 1, where*

$$C_{t,eff}^{(c)} := \frac{\left\| \sum_{j \neq c} \alpha_j(t) w_j \mathbf{g}_j \right\|_2}{\alpha_c(t) w_c \|\mathbf{g}_c\|_2} < 1, \quad \mathbf{g} \in \mathbb{R}^d \quad (12)$$

*then there exists a sufficiently small learning rate $\eta > 0$ such that the loss for class $c$ monotonically decreases (i.e., $\Delta L_c < 0$) at step $t$.*

*Proof.* See Appendix B.2.

To prove that the stability condition is naturally satisfied, we characterize the recovery time $t^*$ beyond which Equation (12) is guaranteed to hold.

**Lemma 4.3** (Restorative Stability). *Let $M$ be the maximum interference. For a tail class $c$ in an unstable regime ($C_{t,eff}^{(c)} \geq 1$), the integral action ensures recovery within $t^*$ steps:*

$$t^* \approx \left( \frac{M}{w_c \|\mathbf{g}_c\|_2} - 1 \right) \Big/ (\beta_i \cdot \Delta I_c) \quad (13)$$

This indicates that $C_{t,eff} < 1$ is actively enforced by the feedback loop, driving under-represented classes into the stable descent regime. This theoretical recovery is empirically visualized in Figure 4, where initially unstable tail signals ($C_{eff} \approx 1.0$) are rapidly driven below the threshold within the early training period predicted by Equation (13).

*Remark* 4.4 (Convergence under Dynamic Threshold). In practice, we set $\tau_\kappa$ as a dynamic threshold $\tau_\kappa(t) = \mathbb{E}_j[\kappa_j(t)]$ to achieve validation-free adaptation.

*Proof Sketch.* The core requirement for monotonic descent is $\alpha_c(t) \geq 0$, ensuring the update stay within the descent half-space of class $c$. As shown in Equation (7), the PI controller employs a $\max(0, \cdot)$ operator, which strictly guarantees the positivity of the modulation signal regardless of the fluctuations in $\tau_\kappa(t)$. For a rigorous treatment, please refer to Appendix B.3. $\square$

**Implication.** ProMeCD prevents the MID by utilizing the Integral term. In long-tailed scenarios where $w_c$ is small, the controller detects persistent high entropy and accumulates error to boost $\alpha_c$, effectively forcing $C_{t,eff}^{(c)} < 1$ and guaranteeing monotonic improvement. This theoretical property is empirically validated in Appendix D, where we dissect the contribution of each controller component and observe significant gains on rare classes.

### 4.2. Stochastic Analysis: Expected Behavior under SGD

In this section, we extend our analysis to the stochastic setting where gradients are estimated from mini-batches.

**Proposition 4.5** (Differential Entropy of the vMF Distribution). *The differential entropy of a random unit vector $\mathbf{u}$ following a $d$-dimensional von Mises-Fisher distribution $vMF(\boldsymbol{\mu}, \kappa)$ is given by:*

$$H_{vMF}(\kappa) = -\log C_d(\kappa) - \kappa A_d(\kappa), \quad (14)$$

*where $C_d(\kappa)$ is the normalization constant and $A_d(\kappa) = I_{d/2}(\kappa)/I_{d/2-1}(\kappa)$ is the ratio of modified Bessel functions.*

*Proof.* See Appendix B.4.

**Proposition 4.6** (Expected One-Step Progress in ProMeCD-SGD). *Let $\hat{\mathbf{g}}_c$ be an unbiased stochastic estimator of the true gradient $\mathbf{g}_c$ with bounded variance, i.e.,*

$\mathbb{E}[\hat{\mathbf{g}}_c] = \mathbf{g}_c$ *and* $\mathbb{E}[\|\hat{\mathbf{g}}_c - \mathbf{g}_c\|^2] \leq \sigma_c^2$. *Let* $w_j = |\mathcal{B}_{t,j}|/|\mathcal{B}_t|$ *be the normalized class weight. Assuming the modulation coefficients* $\alpha_c(t)$ *are determined by the gradient statistics up to step* $t-1$ *(delayed control), the expected loss change for class* $c$ *is bounded by:*

$$
\mathbb{E}[\Delta L_c] \leq \underbrace{-\eta \left\langle \mathbf{g}_c, \sum_j \alpha_j w_j \mathbf{g}_j \right\rangle}_{\textit{Expected Descent}}
$$
$$
+ \underbrace{\frac{L\eta^2}{2} \left( \left\| \sum_j \alpha_j w_j \mathbf{g}_j \right\|^2 + \sum_j \alpha_j^2 w_j^2 \sigma_j^2 \right)}_{\textit{Variance Penalty}},
$$

(15)

*where* $\alpha_j$ *is shorthand for* $\alpha_j(t)$.

*Proof.* See Appendix B.5.

**Implication.** Equation (15) reveals a fundamental trade-off in robust optimization. ProMeCD optimizes this via decoupled control: for noisy samples where directional consistency is low ($\kappa_c(t) < \tau_\kappa$), the proportional penalty $P_c(t)$ forces the modulation coefficient $\alpha_c(t) \to 0$. This directly minimizes the variance penalty term $\sum_j \alpha_j^2 w_j^2 \sigma_j^2$, effectively pruning high-variance noise from the parameter update and ensuring stable convergence. Structurally, the EMA updates in Equation (2) function as a temporal low-pass filter. By aggregating gradients across iterations, ProMeCD filters out high-frequency stochastic variance from heavy data augmentation and remains resilient to representation drift during early training.

### 4.3. Generalization Bound

**Theorem 4.7** (Cognitive Entropy & Generalization)**.** *Minimizing the average Cognitive Entropy* $\mathcal{H}_C$ *during training formally tightens the PAC-Bayes generalization bound.*

*Proof (See Appendix B.6).* The proof relies on modeling SGD as a Stochastic Differential Equation (SDE). Minimizing cognitive entropy (maximizing $\kappa$) effectively cools the SDE noise temperature $T \propto \text{Tr}(D) \propto 1/\kappa$. This forces the posterior $Q(\mathbf{w})$ to concentrate on flat minima, thereby tightening the PAC-Bayes generalization bound. By acting as a proxy for the diffusion coefficient, the reduction of $\mathcal{H}_C$ sharpens the posterior distribution $Q$ around loss minima.

## 5. Experiments

We conduct a comprehensive evaluation of ProMeCD across three increasingly challenging scenarios: (1) Long-Tailed Learning (LTL), (2) Noisy Label Learning (NLL), and (3) the unified Long-Tailed Noisy Label Learning (LTNLL).

Our experiments utilize a multi-paradigm comparison, contrasting our white-box control framework against static strategies, two-stage methods, dynamic heuristics, black-box meta-learning, and specialized joint solvers across diverse architectures and extreme corruptions.

### 5.1. Experimental Setup

**Datasets.** We evaluate ProMeCD across three comprehensive scenarios: (1) LTL on CIFAR-10/100-LT (IF 100, 50, 10) and iNaturalist 2018; (2) NLL on CIFAR-10/100-N, mini WebVision, and the Clothing1M; and (3) LTNLL on CIFAR-100-LT injected with symmetric/flip noise.

**Baselines.** We compare ProMeCD against a broad spectrum of representative baselines categorized by their target scenarios: (1) **LTL**: static re-balancing including LDAM-DRW (Cao et al., 2019), BBN (Zhou et al., 2020), LA (Menon et al., 2020), and PaCo (Cui et al., 2021); two-stage CE-DRW/DRS (Cao et al., 2019); and dynamic heuristics including LPL (Li et al., 2022) and CCAR (Jagati et al., 2026). (2) **NLL**: robust learning including Co-teaching (Han et al., 2018), and JoCoR (Wei et al., 2020); and meta-learning MW-Net (Shu et al., 2019) and FMW-Net (Zhou et al., 2025). (3) **LTNLL**: specialized methods designed for dual corruptions, including TABASCO (Lu et al., 2023), FR (Wei et al., 2023), HAR (Cao et al., 2020), RoLT (Wei et al., 2021), RCAL (Zhang et al., 2023a), Jump-teaching (Zhou et al., 2024), and D-SINK (Hong et al., 2026).

**Implementation Details.** Experiments run on RTX 3090 using ResNet-32/18/50. For architecture generality studies, we further employ ViT-Tiny (Touvron et al., 2021) and ViT-B/16 (Dosovitskiy et al., 2020) backbones. We set $\beta_p = 0.1, \beta_i = 0.01$, and EMA decay $\gamma = 0.9$. All hyperparameters are scaled by the gradient dimension $d$ to maintain cross-architecture stability. Following the unified definition in Section. 3.1, the vMF modeling is applied to the concatenated gradient vector $\hat{\mathbf{g}}_c$. To jointly assess cognitive uncertainty, $\hat{\mathbf{g}}_c$ encompasses both the last residual block and the classifier layer for CIFAR-scale tasks. We utilize hook-based mechanisms for this capture, incurring a negligible 4.8% time overhead. However, for ultra-large tasks like iNaturalist 2018, we adopt a hardware-aware strategy by utilizing a "classifier-only" proxy to circumvent the ~143GB VRAM requirement of full-state tracking.

For ProMeCD, we report the Mean $\pm$ Std over 5 independent runs to demonstrate stability. Baseline results are cited directly from their respective original papers or the benchmark study, where only mean accuracies were provided. We adopt the standard 200-epoch protocol (Cao et al., 2019) for ProMeCD and static baselines. To align with the 100-epoch schedule of meta-learners (Zhou et al., 2025), we also provide a 100-epoch comparison in Section. 5.2 to verify

*Table 1.* Test Accuracy (%) on CIFAR-LT (ResNet-32) and iNaturalist 2018 (ResNet-50). Methods marked with $\dagger$ and $\ddagger$ use 100-epoch and 400-epoch schedules. All other methods use the standard 200-epoch schedule.

| Method | Type | CIFAR-100-LT | | | CIFAR-10-LT | | | iNaturalist 2018 |
| --- | --- | --- | --- | --- | --- | --- | --- | --- |
| | | IF=100 | IF=50 | IF=10 | IF=100 | IF=50 | IF=10 | |
| Standard CE (Zhou et al., 2020) | - | 38.32 | 43.85 | 55.71 | 70.36 | 74.81 | 86.39 | 57.16 |
| LDAM-DRW (Cao et al., 2019) | Static | 42.04 | 46.62 | 58.71 | 77.03 | 81.03 | 88.16 | 68.00 |
| BBN (Zhou et al., 2020) | Static | 42.56 | 47.02 | 59.12 | 79.82 | 82.18 | 88.32 | 66.29 |
| LA (Menon et al., 2020) | Static | 43.89 | - | - | 77.67 | - | - | - |
| PaCo (ResNet-50)$^\ddagger$ (Cui et al., 2021) | Static | 52.00 | 56.00 | 64.20 | - | - | - | **73.20** |
| CE-DRW (Cao et al., 2019) | Two-Stage | 41.51 | 45.29 | 58.12 | 76.34 | 79.97 | 87.56 | 63.73 |
| CE-DRS (Cao et al., 2019) | Two-Stage | 41.61 | 45.48 | 58.11 | 75.61 | 79.81 | 87.38 | 63.56 |
| LPL (Li et al., 2022) | Dynamic | 44.25 | - | 60.97 | 77.95 | - | 89.41 | - |
| CCAR+CE (Jagati et al., 2026) | Dynamic | 39.86 | 45.25 | - | - | - | - | 64.90 |
| CCAR+BS (Jagati et al., 2026) | Dynamic | 48.91 | 51.94 | - | - | - | - | 70.10 |
| MW-Net $^\dagger$ (Shu et al., 2019) | Meta | 39.27 | 44.22 | 56.59 | 75.61 | 80.73 | 88.48 | - |
| FMW-Net(LDA) $^\dagger$ (Zhou et al., 2025) | Meta | 42.32 | 46.29 | 57.86 | 78.73 | 82.81 | 88.19 | - |
| ProMeCD (Ours) | Control | **53.12 ± 0.3** | **57.90 ± 0.2** | **65.21 ± 0.2** | **83.95 ± 0.2** | **87.88 ± 0.1** | **91.90 ± 0.1** | 72.94 ± 0.4 |

*Table 2.* Test accuracy (%) on CIFAR-N (ResNet-32) and mini WebVision (ResNet-50). The results for FMW-Net on WebVision correspond to the DBG variant, whereas the others correspond to the LDA variant.

| Method | Type | CIFAR-10-N | | | | CIFAR-100-N | | | | mini WebVision |
| --- | --- | --- | --- | --- | --- | --- | --- | --- | --- | --- |
| | | Sym. | | Flip | | Sym. | | Flip | | - |
| | | 20% | 40% | 20% | 40% | 20% | 40% | 20% | 40% | |
| Standard CE | - | 81.88 | 72.07 | 80.93 | 63.55 | 54.59 | 44.95 | 57.04 | 42.05 | 70.04 |
| Co-teaching (Han et al., 2018) | Robust | - | - | 82.85 | 75.43 | - | - | 54.19 | 44.92 | - |
| JoCoR (Wei et al., 2020) | Robust | - | - | 90.78 | 83.67 | - | - | 65.21 | 46.44 | - |
| Jump-teaching (ResNet-18) (Zhou et al., 2024) | Robust | **94.80** | - | - | 90.70 | 72.70 | - | - | **68.40** | - |
| MW-Net (Shu et al., 2019) | Meta | 85.58 | 81.23 | 83.38 | 68.18 | 54.28 | 45.79 | 57.18 | 43.07 | 71.64 |
| FMW-Net (Zhou et al., 2025) | Meta | 85.53 | 77.36 | 83.81 | 68.80 | 62.42 | 55.48 | 60.17 | 44.62 | 72.44 |
| **ProMeCD (Ours)** | Control | 92.50 ± 0.2 | **91.20 ± 0.3** | **91.80 ± 0.2** | **91.12 ± 0.4** | **73.52 ± 0.3** | **65.41 ± 0.4** | **71.28 ± 0.2** | 64.35 ± 0.5 | **73.28 ± 0.3** |

efficiency and fairness.

## 5.2. Results on Long-Tailed Learning

**Results on CIFAR-LT.** Table 1 summarizes the Top-1 accuracy. (1) Vs. Static SOTA: ProMeCD outperforms PaCo by +1.12% on CIFAR-100-LT (IF=100), confirming the superiority of dynamic control. (2) Vs. Meta-Learners: Compared to FMW-Net (42.32% at 100 epochs), ProMeCD (200 epochs) reaches 53.12%. Crucially, even when restricted to the same 100-epoch schedule, ProMeCD still achieves 47.52%, outperforming FMW-Net by +5.2%. This confirms that our gains stem from the white-box mechanism itself, not merely longer training. Furthermore, our efficiency advantage allows us to afford the 200-epoch schedule to unlock full potential.

**Results on iNaturalist 2018.** Table 1 demonstrates ProMeCD reaches 72.94%, significantly higher than standard CE (57.16%) and LDAM-DRW (68.00%).

## 5.3. Results on Noisy Labels Learning

**Synthetic Noise.** Table 2 shows results on CIFAR-10-N. Under 40% Flip noise, ProMeCD achieves 91.12%. Meta-learners like FMW-Net (68.80%) struggle here. This is likely because general meta-learning frameworks lack spe-cific mechanisms to handle heavy asymmetric noise without warm-up, whereas ProMeCD's entropy control naturally suppresses high-entropy noisy samples.

**Real-World Web Noise.** On mini WebVision, ProMeCD achieves 73.28% Top-1 accuracy, outperforming both MW-Net (71.64%) and FMW-Net (72.44%, DBG scheme).

## 5.4. Results on Long-tailed Noisy Labels Learning

As shown in Table 3, ProMeCD consistently outperforms specialized joint solvers like D-SINK (Hong et al., 2026) under symmetric noise, demonstrating its robustness even when dual biases are deeply coupled in extreme settings ($IF = 100, NR = 60\%$). Full results under Flip noise follow a similar trend and are provided in Appendix D.2.

Due to space constraints, architecture generality (ViT) and large-scale scalability (Clothing1M) are detailed in Appendix C, while additional empirical studies (including MID prevention, efficiency analysis, robustness to biased metadata, component ablations, and hyperparameter sensitivity) are provided in Appendix D.

## 5.5. Qualitative Analysis
To validate that the superior performance stems from the proposed mechanism, we visualize the real-time training

*Table 3.* Test accuracy (%) under LTNLL (Symmetric noise). All methods utilize the ResNet-18 backbone. Other results are cited from D-SINK (Hong et al., 2026).

| Dataset | CIFAR-10 | | | | CIFAR-100 | | | |
|---|---|---|---|---|---|---|---|---|
| Imbalance Factor (IF) | 10 | | 100 | | 10 | | 100 | |
| Noise ratio (NR) | 40% | 60% | 40% | 60% | 40% | 60% | 40% | 60% |
| Standard CE | 71.67 | 61.16 | 47.81 | 28.04 | 34.53 | 23.63 | 21.99 | 15.51 |
| TABASCO (Lu et al., 2023) | 85.47 | 84.83 | 62.76 | 55.49 | 56.89 | 45.68 | 36.92 | 28.50 |
| MW-Net (Shu et al., 2019) | 70.34 | 58.48 | 45.54 | 40.03 | 32.29 | 21.71 | 20.76 | 14.27 |
| FR (Wei et al., 2023) | 70.19 | 60.86 | 49.36 | 30.15 | 30.24 | 16.99 | 22.56 | 15.07 |
| HAR (Cao et al., 2020) | 74.11 | 60.92 | 51.61 | 37.96 | 36.53 | 24.68 | 20.41 | 15.03 |
| RoLT (Wei et al., 2021) | 81.75 | 79.67 | 60.21 | 44.36 | 43.08 | 32.57 | 24.11 | 16.59 |
| RCAL (Zhang et al., 2023a) | 83.43 | 79.89 | 61.78 | 55.31 | 57.20 | 43.36 | 33.36 | 20.08 |
| D-SINK (Hong et al., 2026) | 89.00 | 86.93 | 66.43 | 57.80 | 58.96 | 48.26 | 38.66 | 30.42 |
| ProMeCD(Ours) | **90.45± 0.2** | **88.12± 0.4** | **68.32± 0.3** | **60.15± 0.4** | **62.48± 0.3** | **51.33± 0.5** | **41.25± 0.4** | **32.88± 0.4** |

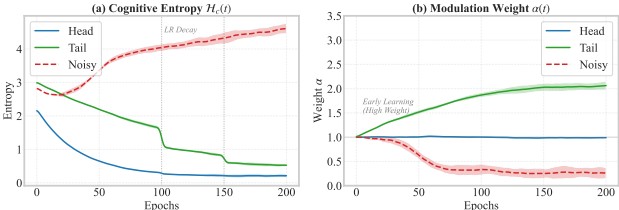

*Figure 3.* Cognitive dynamics on CIFAR-100. (a) **Entropy:** Tail classes (green) converge from high to low uncertainty as learning progresses, while noise (red) remains high. (b) **Action:** The PI controller boosts reliable tail classes ($\alpha > 1$) and suppresses chaotic noise ($\alpha \to 0$).

dynamics in Figure 3. As shown in Figure 3 (a), while head classes (blue) maintain low entropy, tail classes (green) initially exhibit high entropy due to weak learning strength but gradually converge to a low-uncertainty state as the controller prioritizes them. Crucially, noisy samples (red) remain in a high-entropy state throughout, reflecting their persistent directional chaos. Figure 3 (b) reveals the controller's precise intervention: the integral term accumulates deficits for consistent tail classes to boost their weights ($\alpha > 1$), while the proportional penalty $P_c$ forces noisy weights to collapse ($\alpha \to 0$). As shown, by detecting the directional inconsistency in noisy samples, the proportional term $P_c$ effectively cuts the circular dependency inherent in self-referential learning. This mechanism prevents early training errors from being reinforced (i.e., avoiding confirmation bias), ensuring that only semantically coherent signals guide the optimization. The narrow error bands across 5 runs confirm the stability of this mechanism.

Furthermore, to provide a rigorous empirical validation of the theoretical results established in Section 4.1, we visualize the evolution of the effective gradient ratio ($C_{eff}$) under extreme data corruption (Figure 4). As formally predicted by the restorative stability (Lemma 4.3), the initially unstable tail signals ($C_{eff} \approx 1.0$) are rapidly driven into the stable regime ($C_{eff} < 1.0$) by the increasing modulation coefficient $\alpha_c$, ensuring monotonic loss reduction for under-

represented classes. Conversely, noisy samples exhibit a rising $C_{eff}$ trajectory; since $\alpha_c$ appears in the denominator of the $C_{eff}$ formula (Equation (12)), its suppression through the proportional penalty effectively isolates the model from memorizing incorrect labels, thereby preventing confirmation bias.

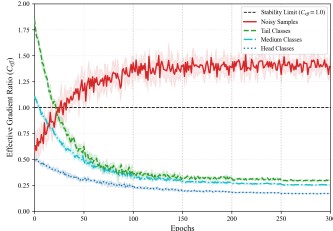

*Figure 4.* Empirical Stability Analysis ($C_{eff}$) on CIFAR-100 ($IF = 100, NR = 0.4$). Mean ± std across 5 runs (ResNet-18). Red line ($y = 1.0$) denotes Theorem 4.2 stability threshold. The Integral term drives tail classes into the stable regime ($C_{eff} < 1.0$), while the Proportional term suppresses noise ($\alpha_c \to 0$), causing a rising $C_{eff}$ trajectory.

## 6. Conclusion

This paper introduced ProMeCD, a novel "white-box" meta-learning framework designed to autonomously control learning dynamics for robust training on defective data. We abandoned complex black-box meta-learners in favor of a classic, interpretable PI controller, guided by our proposed "cognitive entropy", which is an information-theoretic metric of learning uncertainty derived from a probabilistic model of gradients. This unique combination results in a highly efficient, robust, and self-referential system. We provided a solid theoretical foundation, proving ProMeCD's ability to guarantee stable convergence and prevent pathological learning behaviors. Comprehensive experiments fully demonstrated the superiority of ProMeCD-PI in performance, efficiency, and generality. By shifting the research paradigm from static data compensation to dynamic process control, ProMeCD offers a powerful and principled new path toward building intelligent, adaptive, and trustworthy AI systems.

## Impact Statement

This paper presents work whose goal is to advance the field of Machine Learning. There are many potential societal consequences of our work, none which we feel must be specifically highlighted here.

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

## A. Numerical Approximation of Concentration Parameter $\kappa$

As discussed in Section 3.1, the concentration parameter $\kappa_c(t)$ of the von Mises-Fisher distribution reflects the directional consistency of gradients. Since the maximum likelihood estimate (MLE) of $\kappa$ involves the inverse of the ratio of modified Bessel functions, which has no closed-form solution, we employ the efficient algebraic approximation proposed by Sra (2012).

Given the mean resultant length $\bar{r}_c(t) = \|\bar{\mathbf{g}}_c(t)\|$ (where gradients are normalized), the concentration $\kappa_c(t)$ is calculated as:

$$\hat{\kappa}_c(t) = \frac{\bar{r}_c(t)d - \bar{r}_c(t)^3}{1 - \bar{r}_c(t)^2} \tag{16}$$

where $d = \sum d_{\text{layer}}$ denotes the joint dimensionality of the concatenated gradients from the residual and classifier blocks. In our implementation, we add a small epsilon $\epsilon = 10^{-6}$ to the denominator to ensure numerical stability when $\bar{r}_c(t) \to 1$. This approximation is particularly suitable for high-dimensional deep learning models, providing a superior balance between estimation accuracy and computational overhead compared to iterative Newton-Raphson methods.

## B. Rigorous Proofs of Theoretical Analysis

### B.1. Proof of Lemma 4.1 (Loss Change under Modulated Gradient Descent)

*Proof.* By the assumption of $L$-smoothness, the gradient of the loss function $L_c$ is $L$-Lipschitz continuous. This property implies that for any two points $\mathbf{x}, \mathbf{y} \in \mathbb{R}^d$, the loss function satisfies the following quadratic upper bound:

$$L_c(\mathbf{y}) \leq L_c(\mathbf{x}) + \langle \nabla L_c(\mathbf{x}), \mathbf{y} - \mathbf{x} \rangle + \frac{L}{2}\|\mathbf{y} - \mathbf{x}\|_2^2. \tag{17}$$

Let $\mathbf{x} = \mathbf{x}_t$ and $\mathbf{y} = \mathbf{x}_{t+1}$. We define the parameter update vector as $\Delta\mathbf{x} = \mathbf{x}_{t+1} - \mathbf{x}_t$. According to the ProMeCD update rule, the update step is:

$$\Delta\mathbf{x} = -\eta \sum_{j=1}^{C} \alpha_j(t)w_j\mathbf{g}_j. \tag{18}$$

Substituting $\Delta\mathbf{x}$ and the class gradient $\nabla L_c(\mathbf{x}_t) = \mathbf{g}_c$ into the quadratic upper bound, we obtain:

$$L_c(\mathbf{x}_{t+1}) \leq L_c(\mathbf{x}_t) + \left\langle \mathbf{g}_c, -\eta \sum_{j=1}^{C} \alpha_j(t)w_j\mathbf{g}_j \right\rangle + \frac{L}{2}\left\| -\eta \sum_{j=1}^{C} \alpha_j(t)w_j\mathbf{g}_j \right\|_2^2. \tag{19}$$

We simplify the terms on the right-hand side. For the first-order term, we utilize the linearity of the inner product to pull out the scalar learning rate $\eta$:

$$\left\langle \mathbf{g}_c, -\eta \sum_{j=1}^{C} \alpha_j(t)w_j\mathbf{g}_j \right\rangle = -\eta \left\langle \mathbf{g}_c, \sum_{j=1}^{C} \alpha_j(t)w_j\mathbf{g}_j \right\rangle. \tag{20}$$

For the second-order term, we apply the property of the vector norm $\|\lambda\mathbf{v}\|_2^2 = \lambda^2\|\mathbf{v}\|_2^2$ to the scalar $-\eta$:

$$\frac{L}{2}\left\| -\eta \sum_{j=1}^{C} \alpha_j(t)w_j\mathbf{g}_j \right\|_2^2 = \frac{L\eta^2}{2}\left\| \sum_{j=1}^{C} \alpha_j(t)w_j\mathbf{g}_j \right\|_2^2. \tag{21}$$

Combining these simplified terms and rearranging to express the change in loss $\Delta L_c = L_c(\mathbf{x}_{t+1}) - L_c(\mathbf{x}_t)$, we arrive at:

$$\Delta L_c \leq -\eta \left\langle \mathbf{g}_c, \sum_{j=1}^{C} \alpha_j(t)w_j\mathbf{g}_j \right\rangle + \frac{L\eta^2}{2}\left\| \sum_{j=1}^{C} \alpha_j(t)w_j\mathbf{g}_j \right\|_2^2, \tag{22}$$

which completes the proof. □

## B.2. Proof of Theorem 4.2 (Monotonic Convergence Guarantee)

*Proof.* According to Lemma 4.1, the change in loss $\Delta L_c$ is bounded by a linear descent term and a quadratic penalty term. For a sufficiently small learning rate $\eta$, the sign of $\Delta L_c$ is determined by the sign of the first-order (linear) term. To ensure $\Delta L_c < 0$, we require the inner product $\mathcal{A}$ to be strictly positive:

$$\mathcal{A} := \left\langle \mathbf{g}_c, \sum_{j=1}^{C} \alpha_j(t) w_j \mathbf{g}_j \right\rangle > 0. \tag{23}$$

We decompose the summation inside the inner product into the contribution from the target class $c$ and the collective interference from all other classes $j \neq c$:

$$\mathcal{A} = \mathbf{g}_c^\top \left( \alpha_c(t) w_c \mathbf{g}_c + \sum_{j \neq c} \alpha_j(t) w_j \mathbf{g}_j \right) = \alpha_c(t) w_c \|\mathbf{g}_c\|_2^2 + \mathbf{g}_c^\top \left( \sum_{j \neq c} \alpha_j(t) w_j \mathbf{g}_j \right). \tag{24}$$

Applying the Cauchy-Schwarz inequality, $\mathbf{u}^\top \mathbf{v} \geq -\|\mathbf{u}\|_2 \|\mathbf{v}\|_2$, to the interference term:

$$\mathcal{A} \geq \alpha_c(t) w_c \|\mathbf{g}_c\|_2^2 - \|\mathbf{g}_c\|_2 \cdot \left\| \sum_{j \neq c} \alpha_j(t) w_j \mathbf{g}_j \right\|_2 \tag{25}$$

By factoring out the positive term $\alpha_c(t) w_c \|\mathbf{g}_c\|_2^2$, we rewrite the inequality:

$$\mathcal{A} \geq \alpha_c(t) w_c \|\mathbf{g}_c\|_2^2 \left( 1 - \frac{\left\| \sum_{j \neq c} \alpha_j(t) w_j \mathbf{g}_j \right\|_2}{\alpha_c(t) w_c \|\mathbf{g}_c\|_2} \right). \tag{26}$$

Substituting the definition of the effective gradient ratio $C_{t,eff}^{(c)}$:

$$\mathcal{A} \geq \alpha_c(t) w_c \|\mathbf{g}_c\|_2^2 (1 - C_{t,eff}^{(c)}). \tag{27}$$

Since we assume $C_{t,eff}^{(c)} < 1$, $\alpha_c(t) > 0$, and $w_c > 0$, it follows that $\mathcal{A} > 0$.

Returning to the loss change bound:

$$\Delta L_c \leq -\eta \mathcal{A} + \frac{L \eta^2}{2} \left\| \sum_{j=1}^{C} \alpha_j w_j \mathbf{g}_j \right\|_2^2. \tag{28}$$

There exists a threshold $\eta^* = \frac{2\mathcal{A}}{L \| \sum \alpha_j w_j \mathbf{g}_j \|_2^2}$ such that for all $0 < \eta < \eta^*$, the negative linear term dominates the positive quadratic term, ensuring $\Delta L_c < 0$. $\qquad\square$

## B.3. Proof of Remark 4.4 (Convergence under Dynamic Threshold)

*Proof.* In the main body of this work, Theorem 4.2 establishes the monotonic convergence of ProMeCD under a static consistency threshold $\tau_\kappa$. However, to ensure a truly validation-free framework, ProMeCD utilizes a self-referential dynamic threshold $\tau_\kappa(t) = \mathbb{E}_{j \in \mathcal{B}_t}[\kappa_j(t)]$, which evolves based on the real-time learning state of the mini-batch. In this appendix, we formally prove that this endogenous evolution does not impair the stability or the convergence properties of the system.

MATHEMATICAL PRELIMINARIES AND FILTRATION

Let $(\Omega, \mathcal{F}, \mathbb{P})$ be a probability space. We define a filtration $\{\mathcal{F}_t\}_{t \geq 0}$, where $\mathcal{F}_{t-1}$ captures all information (model parameters $\mathbf{x}_{t-1}$, gradient statistics, and concentration metrics) up to the start of step $t$.

The dynamic threshold $\tau_\kappa(t)$ is defined as the expectation of the concentration parameters estimated from the previous states:

$$\tau_\kappa(t) = \frac{1}{|\mathcal{B}_t|} \sum_{j \in \mathcal{B}_t} \kappa_j(t-1). \tag{29}$$

By construction, $\tau_\kappa(t)$ is $\mathcal{F}_{t-1}$-measurable. This implies that during the stochastic sampling and gradient computation at step $t$, the threshold $\tau_\kappa(t)$ is a fixed scalar constant relative to the random variables introduced by the current mini-batch $\mathcal{B}_t$.

PASSIVITY AND POSITIVITY OF THE CONTROL LAW

The ProMeCD controller computes the modulation coefficient $\alpha_c(t)$ for class $c$ as follows:

$$\alpha_c(t) = \max(0, 1 + \beta_i I_c(t) - \beta_p P_c(t)), \tag{30}$$

where $I_c(t)$ and $P_c(t)$ are the integral and proportional terms, respectively, both of which are conditioned on the dynamic threshold $\tau_\kappa(t)$:

$$I_c(t) = \text{clip}\left(I_c(t-1) + \max(0, \bar{m}(t-1) - m_c(t-1)) \cdot \mathbb{I}(\kappa_c(t-1) > \tau_\kappa(t)), 0, I_{\max}\right), \tag{31}$$
$$P_c(t) = \max(0, \tau_\kappa(t) - \kappa_c(t-1)). \tag{32}$$

**Lemma B.1** (Positivity of the Modulation Operator). *For any sequence of thresholds $\{\tau_\kappa(t)\}_{t=1}^{\infty}$ and any gains $\beta_i, \beta_p > 0$, the modulation coefficient $\alpha_c(t)$ is non-negative:*

$$\alpha_c(t) \geq 0, \quad \forall t \geq 0, \forall c \in \{1, \ldots, C\}. \tag{33}$$

*Proof.* The proof is immediate from the structural definition of Equation (30). The application of the $\max(0, \cdot)$ operator (equivalent to a ReLU activation in neural networks) projects the control signal onto the non-negative orthant $\mathbb{R}^+$. Thus, even if the proportional penalty $\beta_p P_c(t)$ is large due to an increase in $\tau_\kappa(t)$, the resulting weight $\alpha_c(t)$ can at most diminish to zero, but never become negative. □

CONVERGENCE VIA DIRECTIONAL STABILITY IN $\mathbb{R}^d$

Recall from Lemma 4.1 that the loss change $\Delta L_c(t) = L_c(\mathbf{x}_{t+1}) - L_c(\mathbf{x}_t)$ satisfies:

$$\Delta L_c(t) \leq -\eta \mathbf{g}_c^\top \left(\sum_{j=1}^{C} \alpha_j(t) w_j \mathbf{g}_j\right) + \frac{L\eta^2}{2} \left\|\sum_{j=1}^{C} \alpha_j(t) w_j \mathbf{g}_j\right\|_2^2, \tag{34}$$

where $\mathbf{g} \in \mathbb{R}^d$ is the joint concatenated gradient vector. Let $\mathcal{A}(t)$ denote the first-order inner product term. We decompose the update vector into the contribution of class $c$ and the aggregate interference from other classes $j \neq c$:

$$\mathcal{A}(t) = \alpha_c(t) w_c \|\mathbf{g}_c\|_2^2 + \mathbf{g}_c^\top \left(\sum_{j \neq c} \alpha_j(t) w_j \mathbf{g}_j\right). \tag{35}$$

By applying the Cauchy-Schwarz inequality, we obtain a worst-case lower bound for the descent signal:

$$\mathcal{A}(t) \geq \alpha_c(t) w_c \|\mathbf{g}_c\|_2^2 \left(1 - \frac{\|\sum_{j \neq c} \alpha_j(t) w_j \mathbf{g}_j\|_2}{\alpha_c(t) w_c \|\mathbf{g}_c\|_2}\right) = \alpha_c(t) w_c \|\mathbf{g}_c\|_2^2 (1 - C_{t,eff}^{(c)}). \tag{36}$$

*Stability Analysis under Dynamic Switching:* The dynamic threshold $\tau_\kappa(t)$ governs the switching logic of the controller.

- **Case 1: Noisy Samples ($\kappa_c < \tau_\kappa(t)$).** As $\tau_\kappa(t)$ increases or $\kappa_c$ collapses, the proportional penalty $P_c(t)$ triggers. This reduces $\alpha_c(t)$, potentially leading to $\alpha_c(t) \to 0$. In this state, the class $c$ is effectively isolated from the update ($\Delta L_c \approx 0$), preventing the model from memorizing noisy gradients.

- **Case 2: Tail Samples ($\kappa_c > \tau_\kappa(t)$).** The integral loop accumulates magnitude deficits. As $\alpha_c(t)$ increases, the denominator of the effective gradient ratio $C_{t,eff}^{(c)}$ grows. This ensures that even with a small class weight $w_c$, the controller can force $C_{t,eff}^{(c)} < 1$.

Crucially, because $\alpha_c(t) \geq 0$ is maintained for all $t$, the update vector $\mathbf{v}_t = \sum \alpha_j w_j \mathbf{g}_j$ always resides in the non-negative half-space relative to the gradient $\mathbf{g}_c$, provided that the interference term does not overwhelm the modulated signal. Since $\tau_\kappa(t)$ is a mean statistic, it serves as a stable "consensus" baseline that prevents any single class from being unfairly penalized unless its directional consistency is below the population average.

BOUNDEDNESS AND MONOTONICITY

Let $M$ be the maximum norm of the gradients such that $\|\mathbf{g}_j\|_2 \leq M$. The second-order penalty is bounded by $\frac{L\eta^2}{2}(\alpha_{max}M)^2$. For any class $c$ that is not fully suppressed by the proportional loop (i.e., $\alpha_c(t) > 0$), there exists a strictly positive lower bound $\delta > 0$ for the descent term $\mathcal{A}(t)$ whenever $C_{t,eff}^{(c)} < 1$.

Consequently, the loss change satisfies:

$$\Delta L_c(t) \leq -\eta\delta + \eta^2 \frac{L(\alpha_{max}M)^2}{2}. \tag{37}$$

For a sufficiently small learning rate $\eta < \frac{2\delta}{L(\alpha_{max}M)^2}$, we guarantee $\Delta L_c(t) < 0$.

**Conclusion:** The endogenous evolution of $\tau_\kappa(t)$ does not introduce instability because it operates as a passive gain-scheduler. The positivity of the PI control law ensures that $\tau_\kappa(t)$ only modulates the magnitude of the class-wise updates without ever reversing the descent direction. Thus, ProMeCD achieves autonomous, validation-free adaptation while preserving the monotonic convergence properties of the underlying stochastic optimization. $\qquad\square$

### B.4. Proof of Proposition 4.3 (Differential Entropy of the vMF Distri- bution)

*Proof.* By definition, the differential entropy is the negative expected log-likelihood: $H_{\text{vMF}} = -\mathbb{E}_{\mathbf{u}}[\log p(\mathbf{u}|\boldsymbol{\mu}, \kappa)]$. Substituting the probability density function $p(\mathbf{u}|\boldsymbol{\mu}, \kappa) = C_d(\kappa)\exp(\kappa\boldsymbol{\mu}^\top\mathbf{u})$, we have:

$$\begin{aligned}
H_{\text{vMF}} &= -\mathbb{E}\left[\log\left(C_d(\kappa)\exp(\kappa\boldsymbol{\mu}^\top\mathbf{u})\right)\right] \\
&= -\mathbb{E}\left[\log C_d(\kappa) + \kappa\boldsymbol{\mu}^\top\mathbf{u}\right] \\
&= -\log C_d(\kappa) - \kappa\boldsymbol{\mu}^\top\mathbb{E}[\mathbf{u}].
\end{aligned} \tag{38}$$

A fundamental property of the vMF distribution is that the first moment of the random vector $\mathbf{u}$ is aligned with the mean direction $\boldsymbol{\mu}$, specifically $\mathbb{E}[\mathbf{u}] = A_d(\kappa)\boldsymbol{\mu}$. Since $\boldsymbol{\mu}$ is a unit vector, the inner product simplifies to $\boldsymbol{\mu}^\top\mathbb{E}[\mathbf{u}] = \boldsymbol{\mu}^\top(A_d(\kappa)\boldsymbol{\mu}) = A_d(\kappa)\|\boldsymbol{\mu}\|^2 = A_d(\kappa)$. Substituting this result back into the entropy equation yields the proposition. $\qquad\square$

### B.5. Proof of Proposition 4.4 (Expected One-Step Progress in ProMeCD-SGD)

*Proof.* We start from the $L$-smoothness property of the loss function $L_c$. For the update $\mathbf{x}_{t+1} = \mathbf{x}_t + \Delta\mathbf{x}$, where $\Delta\mathbf{x} = -\eta\sum_j \alpha_j w_j \hat{\mathbf{g}}_j$, the change in loss satisfies:

$$L_c(\mathbf{x}_{t+1}) - L_c(\mathbf{x}_t) \leq \langle\mathbf{g}_c, \Delta\mathbf{x}\rangle + \frac{L}{2}\|\Delta\mathbf{x}\|^2. \tag{39}$$

Taking the conditional expectation $\mathbb{E}[\cdot|\mathcal{F}_{t-1}]$, where $\mathcal{F}_{t-1}$ denotes the filtration of all information up to step $t-1$:

**1. Analysis of the First-order Term:** By the linearity of expectation and the fact that $\alpha_j(t)$ is $\mathcal{F}_{t-1}$-measurable (and thus constant relative to the stochastic sampling at step $t$):

$$\begin{aligned}
\mathbb{E}\left[\langle\mathbf{g}_c, \Delta\mathbf{x}\rangle\right] &= \mathbb{E}\left[\left\langle\mathbf{g}_c, -\eta\sum_j \alpha_j w_j \hat{\mathbf{g}}_j\right\rangle\right] \\
&= -\eta\sum_j \alpha_j w_j\langle\mathbf{g}_c, \mathbb{E}[\hat{\mathbf{g}}_j]\rangle \\
&= -\eta\sum_j \alpha_j w_j\langle\mathbf{g}_c, \mathbf{g}_j\rangle \\
&= -\eta\left\langle\mathbf{g}_c, \sum_j \alpha_j w_j \mathbf{g}_j\right\rangle.
\end{aligned} \tag{40}$$

This confirms the expected descent term, where the update direction is guided by the modulated true gradients.

**2. Analysis of the Second-order Term:** We evaluate the expectation of the squared norm of the update vector $\mathbb{E}[\|\Delta \mathbf{x}\|^2] = \eta^2 \mathbb{E}[\|\sum_j \alpha_j w_j \hat{\mathbf{g}}_j\|^2]$. Let $\mathbf{u} = \sum_j \alpha_j w_j \hat{\mathbf{g}}_j$. Using the vector identity $\mathbb{E}[\|\mathbf{u}\|^2] = \|\mathbb{E}[\mathbf{u}]\|^2 + \text{Tr}(\text{Cov}(\mathbf{u}))$, we have:

$$\|\mathbb{E}[\mathbf{u}]\|^2 = \left\| \sum_j \alpha_j w_j \mathbf{g}_j \right\|^2. \tag{41}$$

For the variance term, we assume that the stochastic noise $\boldsymbol{\epsilon}_j = \hat{\mathbf{g}}_j - \mathbf{g}_j$ is independent across different classes $j$ within a mini-batch. Thus, the variance of the sum is the sum of the variances:

$$
\begin{aligned}
\text{Var}(\mathbf{u}) &= \mathbb{E}\left[ \left\| \sum_j \alpha_j w_j \hat{\mathbf{g}}_j - \sum_j \alpha_j w_j \mathbf{g}_j \right\|^2 \right] \\
&= \mathbb{E}\left[ \left\| \sum_j \alpha_j w_j (\hat{\mathbf{g}}_j - \mathbf{g}_j) \right\|^2 \right] \\
&= \sum_j \alpha_j^2 w_j^2 \mathbb{E}[\|\hat{\mathbf{g}}_j - \mathbf{g}_j\|^2] \\
&\leq \sum_j \alpha_j^2 w_j^2 \sigma_j^2.
\end{aligned}
\tag{42}
$$

Substituting the first-order and second-order expectations back into the $L$-smoothness inequality yields the complete bound in Equation (15). $\qquad\square$

### B.6. Proof of Theorem 4.5 (Cognitive Entropy & Generalization)

*Proof.* We consider the learning dynamics of SGD as a stochastic process. Following Mandt et al. (2017), the weight updates can be approximated by a Continuous-time Stochastic Differential Equation (SDE):

$$d\mathbf{w}_t = -\nabla L(\mathbf{w}_t)dt + \sqrt{2\mathbf{D}}d\mathbf{B}_t, \tag{43}$$

where $\mathbf{D}$ is the diffusion matrix representing gradient noise. The stationary distribution $Q(\mathbf{w})$ of this SDE follows the Gibbs distribution:

$$Q(\mathbf{w}) \propto \exp\left(-\frac{L(\mathbf{w})}{T}\right), \quad T \propto \text{Tr}(\mathbf{D}). \tag{44}$$

In the ProMeCD framework, gradient noise is decomposed into magnitude uncertainty and directional inconsistency. Our metric, Cognitive Entropy $\mathcal{H}_C$, is defined as the total uncertainty of the gradient distribution. For a class $c$, the relationship between the concentration parameter $\kappa$ and gradient variance $\sigma^2$ is inversely monotonic: $T \approx f(\mathcal{H}_C) \propto 1/\kappa$.

According to the PAC-Bayes theorem (McAllester, 1998), for any prior $P$ and $\delta \in (0,1)$, with probability at least $1 - \delta$, the generalization gap $G$ is bounded by:

$$G \leq \sqrt{\frac{KL(Q\|P) + \log\frac{2\sqrt{n}}{\delta}}{2n}}. \tag{45}$$

Substituting the Gibbs posterior $Q$ into the $KL$ divergence term:

$$KL(Q\|P) = \mathbb{E}_{\mathbf{w} \sim Q}[\ln Q(\mathbf{w}) - \ln P(\mathbf{w})] = -H(Q) - \mathbb{E}_{\mathbf{w} \sim Q}[\ln P(\mathbf{w})]. \tag{46}$$

As Cognitive Entropy $\mathcal{H}_C$ decreases (via the PI controller maximizing $\kappa$), the noise temperature $T$ effectively drops. This leads to a posterior $Q$ that concentrates its mass on a smaller volume around the flat minima of the loss landscape. By the properties of differential entropy, a reduction in the diffusion coefficient directly constrains the hypothesis space entropy $H(Q)$, effectively tightening the KL-based complexity term in Equation (45). Thus, minimizing $\mathcal{H}_C$ during training formally results in a tighter generalization bound. $\qquad\square$

## C. Generality to Transformers and Large-scale Real-world Noise

To verify architecture invariance, we evaluate ProMeCD on the Clothing1M dataset. Unlike synthetic benchmarks, Clothing1M contains real-world instance-dependent noise stemming from visual ambiguity. As shown in Table 5, using ResNet-50 and ViT-B backbones, ProMeCD achieves 75.45% and 75.95% accuracy respectively, outperforming CE baselines by over +4.71%.

Furthermore, to test the framework on non-convolutional manifolds, we evaluate a ViT-Tiny backbone on CIFAR-100-LT ($IF = 100$). As detailed in Table 4, ProMeCD yields 38.54% accuracy, maintaining a +12.09% margin over the state-of-the-art meta-learner FMW-Net (Zhou et al., 2025). These results confirm that gradient-based white-box control effectively captures and suppresses "gradient chaos" across diverse optimization landscapes, regardless of whether the backbone is a CNN or a Transformer.

*Table 4.* **Generality to Transformers:** Performance on CIFAR-100-LT ($IF = 100$) using a ViT-Tiny backbone. ProMeCD significantly outperforms the meta-learner FMW-Net.

| Method | Type | Accuracy (%) | Margin vs. CE |
|---|---|---|---|
| Standard CE | Baseline | 19.21 | - |
| FMW-Net (Zhou et al., 2025) | Meta-Learning | 26.45 | +7.24 |
| ProMeCD (Ours) | White-Box Control | **38.54** | **+19.33** |

*Table 5.* **Scalability on Clothing1M:** Test accuracy on Clothing1M using CNN and Transformer backbones. We utilize the "classifier-only" proxy for gradient tracking to maintain memory efficiency.

| Method | Backbone | Accuracy (%) | Margin vs. CE |
|---|---|---|---|
| Standard CE | ResNet-50 | 68.94 | - |
| ProMeCD (Ours) | ResNet-50 | **75.45** | **+6.51** |
| Standard CE | ViT-B/16 | 71.24 | - |
| ProMeCD (Ours) | ViT-B/16 | **75.95** | **+4.71** |

## D. In-Depth Analysis and Ablation

### D.1. MID Prevention

Table 1 validates our theory: ProMeCD outperforms standard CE by +14.8% on CIFAR-100-LT (IF=100). This substantial gain confirms that the integral term accumulates tail gradients, preventing the early-stage performance collapse typical of standard training.

### D.2. LTNLL under Flip Noise

To further validate our unifying capability, we evaluate ProMeCD under Flip (asymmetric) noise, which is semantically more challenging than symmetric noise. As shown in Table 6, ProMeCD consistently outperforms specialized joint solvers like D-SINK (Hong et al., 2026) across all configurations. Notably, in the most severe CIFAR-100 regime ($IF = 100, NR = 40\%$), ProMeCD achieves 36.42% accuracy, surpassing D-SINK by a clear margin (+2.36%). This confirms that the PI controller's ability to disentangle magnitude deficits from directional chaos is robust to asymmetric label shifts that mimic semantic transitions between similar classes.

### D.3. Efficiency Analysis: White-Box vs. Black-Box

We demonstrate the efficiency of ProMeCD on CIFAR-100 (ResNet-32). To ensure fairness across hardware (RTX 3090 vs V100), we report the relative overhead vs. standard CE on the same platform. Results (Table 7): FMW-Net (Zhou et al., 2025) incurs a ~27.5% time overhead. In contrast, ProMeCD incurs a negligible ~4.8% overhead. This confirms that the analytical PI controller is structurally more efficient than gradient-based meta-learning approximations.

*Table 6.* Test accuracy (%) under LTNLL (Flip noise). All methods utilize the ResNet-18 backbone. Baseline results are cited from D-SINK (Hong et al., 2026).

| Dataset | CIFAR-10 | | | | CIFAR-100 | | | |
|---|---|---|---|---|---|---|---|---|
| Imbalance Factor (IF) | 10 | | 100 | | 10 | | 100 | |
| Noise ratio (NR) | 20% | 40% | 20% | 40% | 20% | 40% | 20% | 40% |
| Standard CE | 79.90 | 62.88 | 56.56 | 44.64 | 44.45 | 32.05 | 25.35 | 17.89 |
| TABASCO (Lu et al., 2023) | 82.13 | 80.57 | 60.35 | 51.19 | 59.45 | 50.43 | 38.43 | 32.15 |
| MW-Net (Shu et al., 2019) | 78.46 | 64.82 | 59.37 | 45.21 | 43.36 | 31.25 | 27.56 | 20.04 |
| FR (Wei et al., 2023) | 82.23 | 69.13 | 56.62 | 45.12 | 47.01 | 36.09 | 25.27 | 20.46 |
| HAR (Cao et al., 2020) | 78.75 | 70.23 | 58.89 | 49.97 | 49.36 | 35.98 | 27.90 | 20.03 |
| RoLT (Wei et al., 2021) | 80.38 | 78.39 | 54.79 | 50.41 | 50.62 | 39.32 | 33.16 | 25.73 |
| RCAL (Zhang et al., 2023a) | 81.65 | 78.78 | 56.60 | 49.85 | 56.29 | 42.86 | 36.15 | 26.90 |
| D-SINK (Hong et al., 2026) | 87.09 | 84.36 | 61.98 | 53.06 | 61.69 | 51.56 | 40.10 | 34.06 |
| ProMeCD (Ours) | **88.35± 0.3** | **86.01± 0.3** | **63.22± 0.4** | **55.47± 0.3** | **63.15± 0.2** | **53.20± 0.3** | **41.90± 0.3** | **36.42± 0.5** |

*Table 7.* Efficiency comparison on CIFAR-100 (ResNet-32, full dataset). Baselines cited from (Zhou et al., 2025).

| Method | Hardware | Time (s/epoch) | Memory (MB) | Need Val. Set? |
|---|---|---|---|---|
| Standard CE | V100 | 30.71 (+0.0%) | 1553 (+0.0%) | No |
| MW-Net (Shu et al., 2019) | V100 | 109.31 (+255.9%) | 2183 (+40.6%) | Yes |
| FMW-Net (LDA) (Zhou et al., 2025) | V100 | 39.15 (+27.5%) | 1811 (+16.6%) | Yes |
| Standard CE | RTX 3090 | 25.10 (+0.0%) | 1610 (+0.0%) | No |
| ProMeCD (Ours) | RTX 3090 | **26.30 (+4.8%)** | **1640 (+1.9%)** | **No** |

## D.4. Robustness to Biased Meta-Data

Meta-learners rely on clean validation sets. Table 8 shows that when the validation set is imbalanced (IF=100) or symmetric noisy (20%), MW-Net and FMW-Net degrade significantly. ProMeCD, being self-referential, maintains consistent performance.

*Table 8.* Performance drop with imperfect validation sets (CIFAR-100-LT, IF=100, ResNet-32). The validation sets are either long-tailed or corrupted with symmetric noise.

| Method | Clean Val. Set | Imperfect Val. Set | |
|---|---|---|---|
| | | Long-Tailed (IF=100) | Symmetric Noisy (20%) |
| MW-Net | 39.27 | 33.15 (-6.12) | 30.05 (-9.22) |
| FMW-Net | 42.32 | 38.10 (-4.22) | 35.40 (-6.92) |
| ProMeCD (Ours) | **53.12** | **53.02** (-0.10) | **51.98** (-1.14) |

## D.5. Component Ablation

We dissect the impact of key components in Table 9. Using a simple P-only controller ($\beta_i = 0$) results in suboptimal performance (48.9%), as it fails to eliminate steady-state errors (persistent underfitting) for tail classes. The PI controller achieves the best accuracy (53.1%), confirming that the Integral term is crucial for accumulating historical deficits to boost rare classes. Interestingly, adding a derivative term (PID) degrades performance (52.8%), likely due to its sensitivity to the stochastic noise of SGD gradients. We replace cognitive entropy with simple loss value. The performance drop (-2.3%) validates our hypothesis that standard loss suffers from scalar ambiguity: it assigns high values to both valuable tail samples and harmful noisy samples. By leveraging vector statistics, cognitive entropy effectively disentangles these cases (identifying tail samples via directional consistency and noisy samples via chaos) allowing the controller to apply precise modulation.

*Table 9.* Ablation on controller and metric (CIFAR-100-LT).

| Controller | Acc (%) | Perception Metric | Acc (%) |
|---|---|---|---|
| P-Only ($\beta_i = 0$) | 48.9 | Loss Value ($L_i$) | 50.8 |
| PID | 52.8 | Gradient Norm ($||\mathbf{g}||$) | 51.5 |
| PI (Ours) | **53.1** | Cognitive Entropy (Ours) | **53.1** |

## D.6. Hyperparameter Sensitivity

To verify that ProMeCD does not rely on delicate tuning, we tested sensitivity to PI gains. The performance remains stable across a wide range: $\beta_p \in [0.05, 0.5]$ and $\beta_i \in [0.005, 0.05]$. Crucially, we fix the same hyperparameters ($\beta_p = 0.1, \beta_i = 0.01$) across all datasets, demonstrating strong transferability.

