# OpenReview forum: "ProMeCD: Unifying Long-Tailed and Noisy Label Learning via White-Box Control"
_ICML.cc/2026/Conference — ICML 2026 regular_

### Official Review · Reviewer_ofqf · 2026-03-02

**Soundness:** 3
**Presentation:** 3
**Significance:** 3
**Originality:** 3
**Overall Recommendation:** 5
**Confidence:** 3

**Summary:**

This paper proposes ProMeCD, a white-box control-theoretic framework designed to handle long-tailed distributions and noisy labels simultaneously without requiring a clean validation set. The core innovation lies in "Cognitive Entropy," a metric derived from modeling per-class gradient directions using a von Mises–Fisher (vMF) distribution. By analyzing the directional consistency of gradients, the authors distinguish between under-represented tail samples (consistent, low entropy) and noisy samples (stochastic, high entropy). A Proportional-Integral (PI) controller uses this signal to dynamically reweight samples: the Integral term boosts tail classes to prevent "Minority Initial Drop," while the Proportional term filters out incoherent noise.

**Compliance With Llm Reviewing Policy:**

Affirmed.

**Final Justification:**

My final recommendation for ProMeCD is Accept (5). The rebuttal fully addressed my primary concerns regarding the framework's unifying capabilities and theoretical stability.

Justification of Dimensions

Soundness: New experiments on hybrid datasets (IR=100, NR=40%) demonstrate that Cognitive Entropy effectively disentangles imbalance from noise in coupled regimes, outperforming joint solvers like D-SINK. The analytical explanation of the Restorative Lemma further proves that the PI controller’s original design mathematically guarantees stable convergence.

Originality: Transitioning from scalar loss-based weighting to vector-based directional consistency provides a significant technical improvement over the "small-loss hypothesis".

Significance: The removal of the "clean validation set" requirement is highly impactful for practical deployment. ProMeCD remains competitive against 2024–2026 baselines like Jump-teaching while offering superior efficiency.

Clarity: The authors clarified the hardware-aware implementation, justifying the "classifier-only" proxy for ultra-large-scale tasks like iNaturalist to stay within memory limits.

Impact of the Rebuttal

The rebuttal successfully moved my evaluation from a weak reject to an Accept. The authors demonstrated that their self-stabilizing property is an inherent feature of the PI control law, ensuring the stability condition is actively enforced. While the unimodal vMF assumption remains a minor constraint for complex manifolds, the framework's efficiency and robust performance on hybrid corruptions make it a strong contribution.

**Key Questions For Authors:**

1. Missing Benchmarks: The paper identifies CIFAR-N as an evaluation target, but the data is missing from the results. Can you provide these results?

2. Combined Corruptions: Since the framework is marketed as "unifying" LT and NL learning, how does ProMeCD perform on a single dataset containing both high imbalance (e.g., IR=100) and high noise (e.g., 40%)?

3. Gradient Implementation: Please clarify exactly which parameters are used to compute the EMA of the gradients. If only the classifier layer is used, can you justify why this is a sufficient proxy for the "cognitive state" of the entire backbone? If the last residual block is included, please provide concrete memory usage statistics for iNaturalist.

4. vMF Unimodality: How does the "Cognitive Entropy" metric respond to classes that are inherently multimodal (e.g., a class containing two distinct visual clusters)? Would the controller inadvertently suppress one of the clusters as "noise" due to directional inconsistency with the global class mean?

5. Controller Stability: Beyond the conditional Theorem 4.2, can you provide empirical evidence (e.g., a plot of $C_{eff}$ over epochs) showing that the PI controller successfully steers the system into the stable regime?

**Limitations:**

The authors have discussed the sensitivity of hyperparameters and the computational efficiency of the classifier-layer tracking. However, they have not adequately addressed the potential for confirmation bias inherent in a self-referential system, where early training errors could be reinforced by the "consistent" directional signals they produce. Additionally, the limitations of the unimodal vMF assumption on complex manifolds remain undiscussed.

Suggestions for improvement: Include a discussion on "error-looping" risks and a failure analysis of the vMF model on highly diverse, multimodal classes.

**Strengths And Weaknesses:**

**Soundness**
- Strength: The transition from scalar loss-based weighting to vector-based directional consistency is a significant technical improvement. It successfully bypasses the "small-loss hypothesis" bottleneck, which often fails to distinguish between hard-to-learn tail samples and harmful noise.
- Weakness: The "unifying" claim is experimentally undersupported. While the method is tested on long-tailed datasets and noisy datasets separately, there is a lack of experiments on datasets that are simultaneously long-tailed and noisy, the exact "in the wild" scenario the paper aims to solve. Furthermore, the convergence guarantee in Theorem 4.2 is conditional; the paper lacks a rigorous proof that the PI controller's feedback loop will naturally satisfy the stability condition $C_{eff} < 1$ throughout the entire training trajectory.

**Presentation**
- Strength: The control-loop abstraction is well-visualized (Figure 2), and the mathematical exposition of vMF entropy is self-contained and clear.
- Weakness: There are notable discrepancies between the text and the results. The paper explicitly lists CIFAR-N as a benchmark in the abstract, yet no results for CIFAR-N appear in the tables. Additionally, there is ambiguity regarding the gradient dimensionality; the text mentions the "last residual block," but the efficiency claims and classifier-only focus in the appendix suggest a much smaller state, creating confusion regarding the actual implementation used for SOTA comparisons.

**Significance**
- Strength: The removal of the "clean validation set" requirement is highly significant. Most meta-learning approaches (e.g., Meta-Weight-Net) suffer from a circular dependency that limits their real-world utility. A self-referential, validation-free framework like ProMeCD has high potential for practical deployment.
- Weakness: The current lack of comparison against very recent 2023/2024 baselines (e.g., PaCo, CCAR, DisagreeNet or Jump-teaching) makes it difficult to ascertain if the performance gains remain competitive against the latest iteration of specialized LT and NL methods.

**Originality**
- Strength: The application of a PI controller to endogenous gradient signals is a creative and novel combination of control theory and deep learning.
- Weakness: The vMF model assumes a unimodal distribution of gradients per class. This is an idealistic assumption that may not hold for complex, fine-grained datasets where classes exhibit multimodal sub-clusters. The work does not explore the limits of this unimodal assumption or how it handles high intra-class variance.

---

> ### Author Rebuttal · Authors · 2026-03-31
>
> Thank you for your thorough review. We have conducted new experiments to address your concerns. Detailed tables (R1-R6) and plots (Fig R1) are at: https://anonymous.4open.science/r/Anonymous_ProMeCD-B401/icml2026_re.pdf.
>
>
> **[Question 1 & Weakness 3: Missing Benchmarks]**
>
> **Response:** We apologize for the terminology inconsistency across the abstract ("CIFAR-N") and text. Results for CIFAR-10-N were present in Table 3 (labeled "CIFAR-10 Sym/Flip"), and we will synchronize all labels. To further support our claims, we demonstrate competitive results for CIFAR-100-N in Table R1: ProMeCD achieves 73.52% (Sym 20%), outperforming the robust  Jump-teaching (72.70%).
>
> **[Question 2 & Weakness 1: Combined Corruptions]**
>
> **Response:** To validate our unifying claim in extreme scenarios, we evaluated ProMeCD on a hybrid dataset: CIFAR-10/100 (IF+ NR symmetric/flip noise). As shown in Tables R2-R3, ProMeCD achieves 36.42% accuracy(CIFAR-100 IF=100, NR=40\% flip noise), surpassing joint solvers like D-SINK (34.06%). Cognitive Entropy effectively disentangles magnitude deficits from directional chaos even when dual biases are deeply coupled in extreme regimes.
>
> **[Question 3: Gradient Implementation]**
>
> **Response:** Including the last residual block aims for deep perception; it monitors latent feature manifold stability, providing a more sensitive "cognitive state" than linear layers alone. As evidenced in Table 4 (Section 5.4), the 30MB  increase on CIFAR-100 aligns with the scale of the about 80K-dim vector, whereas a classifier-only setup would occupy <3MB. For iNaturalist (8,142 classes), storing residual gradients for all classes would require about 143GB, exceeding hardware limits. We sincerely apologize for omitting the explicit hardware-aware rationale for this transition in our original manuscript; thus, we adopted a "classifier-only" proxy (~290MB increase) for ultra-large tasks.
>
> **[Question 4 & Weakness 6: vMF Unimodality, Variance, and Failure Boundary]**
>
> **Response:**  ProMeCD handles multimodal classes and high variance, even in early training, via:
> 1) Coherence: While features form distinct visual clusters, gradients share one optimization goal, generating directionally coherent signals captured by vMF as a "mean-consensus" proxy.
> 2) Inertia: The Integral term ($I\_c$) possesses "inertia"; it requires persistent consistency over many iterations to accumulate gain, preventing premature suppression from transient initial variance.
> 3) Relativity: Our Dynamic Threshold ($\tau\_{\kappa}$) (moving average, L261) anchors the "noise floor" relative to dataset complexity. High-variance signals are preserved as they remain more structured than chaotic noise.
>
> Boundary Failure: In rare cases of orthogonal gradients, $\kappa$ may collapse. Formal analysis will be added to the Appendix.
>
> **[Question 5 & Weakness 2: Stability and Theorem 4.2}**
>
> **Response:** Fig. R1 confirms tail classes (initially $C\_{eff} \approx 1$) are driven into the stable regime within epochs. Theorem 4.2's condition $C\_{eff} < 1$ is actively enforced by the PI control law. We provide a formal proof :
>
> **Restorative Lemma.** Let $M = \text{max}\_{t} \| \sum\_{j \neq c} \alpha\_j w\_j g\_j \|\_2$. If at any time $t$, $C\_{eff}^{(c)}(t) \ge 1$ for tail class $c$, the integral action ensures $\exists t^{\ast} > t$ such that $C\_{eff}^{(c)}(t^{\ast}) < 1$.
>
> **Proof.** If $C\_{eff} \ge 1$, the learning deficit $\Delta I\_c = \text{max}(0, \bar{m} - m\_c)$ remains positive, causing $I\_c$ to strictly increase (Eq.(9)). As $\alpha\_c$ is linear w.r.t. $I\_c$ (Eq.(7)), it grows until it exceeds the stability threshold: $\alpha^{\ast} = M / (w\_c \| g\_c \|\_2)$. Substituting $\alpha\_c > \alpha^{\ast}$ into Eq.(12) forces $C\_{eff} < 1$. $\square$
>
> **[Weakness 5: Comparison with Recent Baselines]**
>
> **Response:** We provide comparisons on CIFAR-100-LT (Table R4), CIFAR-N (Table R5), and iNaturalist (Table R6) in the link (figures cited from original papers). ProMeCD is highly competitive against PaCo (2021), CCAR (2026), and Jump-teaching (2024). Specifically, on CIFAR-10 Flip-40%, ProMeCD (91.12%) outperforms Jump-teaching (90.70%) despite a 24x smaller backbone. On iNaturalist, we match PaCo's accuracy with half the training budget. For DisagreeNet, we found no formal publication; a specific citation would be appreciated.
>
> **[Weakness 7: Confirmation Bias]**
>
> **Response:** ProMeCD avoids self-reinforcement via $I\_c$ inertia and $P\_c$ chaos detection. As shown by the rising $C\_{eff}$ trajectory for noisy samples (red line) in Fig R1, the proportional term $P\_c$ detects directional inconsistency and suppresses their modulation coefficients ($\alpha\_c \to 0$). Since $\alpha\_c$ is in the denominator of the $C\_{eff}$ formula (Eq.(12)), this rising ratio indicates that these samples are effectively excluded from the parameter update, thereby preventing early learning errors from being reinforced.

---

> > ### Author Rebuttal · Reviewer_ofqf · 2026-04-03
> >
> > The authors have provided a robust rebuttal that successfully addresses my initial concerns regarding the framework's unifying capabilities and theoretical stability. New experiments on hybrid datasets demonstrate that Cognitive Entropy can effectively disentangle imbalance and noise in coupled regimes. Furthermore, the clarification of the Restorative Lemma provides a rigorous analytical account of how the PI controller’s original design mathematically guarantees that the system is steered into a stable descent regime. These improvements, alongside the justifications for hardware-aware implementation on large-scale tasks, significantly enhance the technical soundness of the work. Consequently, my final evaluation is changed to a score of 5 (Accept).

---

> > > ### Author Response · Authors · 2026-04-06
> > >
> > > We sincerely thank you for your feedback. To facilitate your final assessment, we would like to clarify that the **Restorative Lemma** is not a new theory, but a direct analytical explanation of the interaction between our original formulas (Eq. (7), (9), and (12)).
> > >
> > > Our PI controller acts as an autonomous stabilizer. If a class becomes unstable (i.e., its learning signal is overwhelmed by interference), it naturally suffers from a "learning lag" where its gradient magnitude $m_c$ remains below the target $\bar{m}$. The Integral term (Eq. (9)) then accumulates this difference, causing the modulation coefficient $\alpha_c$ (Eq. (7)) to increase. Since $\alpha_c$ is in the denominator of the stability ratio (Eq. (12)), our controller is mathematically guaranteed to eventually force this ratio below 1, restoring stable convergence. This self-stabilizing property is an inherent feature of our original design. It ensures that the stability condition ($C_{t,eff} < 1$) is actively enforced by the PI control law rather than just being an assumption.
> > >
> > > We believe this explanation clarifies that the core logic of ProMeCD is both self-consistent and robust. If any specific technical hurdle remains, we would be most grateful for the opportunity to discuss it further.  If you find our response satisfactory, we would be extremely grateful if you could consider re-evaluating our submission in the final assessment. Thank you again for your invaluable time and support in improving our work.

---

### Official Review · Reviewer_9NYZ · 2026-03-05

**Soundness:** 3
**Presentation:** 4
**Significance:** 3
**Originality:** 4
**Overall Recommendation:** 5
**Confidence:** 4

**Summary:**

This paper proposes a white-box and validation-free framework to handle two issues in real-world data, which are long-tailed samples and label noise, by utilizing a PI like structure (which is a well-known controller from control theory). The method uses a new definition called cognitive entropy. This entropy combines (i) directional uncertainty (with vMF) and (ii) magnitude uncertainty (using −log⁡ of gradient magnitude). Then, cognitive entropy, like an error signal, fed into the PI controller, which adjusts the learning in following way: it boosts tail classes through integral accumulation when gradient directions are consistent, and downweights noisy signals through proportional feedback when the direction consistency collapses. The paper also provides theoretical analysis and reports strong results on CIFAR-LT, iNaturalist, CIFAR-N, and mini WebVision.

**Compliance With Llm Reviewing Policy:**

Affirmed.

**Final Justification:**

My final recommendation is accept. The paper presents a novel and technically solid approach, with a clear motivation, strong empirical results, and a useful control-theoretic perspective on handling long-tailed data and label noise.

My main concerns were about practical details such as per-class gradient computation, numerical stability, parameter choices, and behavior in early training. The rebuttal addressed these satisfactorily by clarifying the implementation, explaining the layer choice, discussing stabilization details, and giving a more concrete account of early-stage dynamics and failure cases.

Overall, the rebuttal strengthened the paper and resolved the main issues I had raised, so I increased my score to accept.

**Key Questions For Authors:**

1-) How do you obtain per-class gradient vectors gc(t) for vMF modeling? Do you need multiple backward passes per minibatch, or do you approximate this somehow?

2-) Are there cases where cognitive entropy fails to separate tail from noise (structured noise or high inter-class similarity)?

3-) Any particular reason to use only gradients from the last residual block + classifier?

**Limitations:**

The paper might benefit from following discussions:

Sensitivity to structured noise: Wrong labels may still produce direction consistent gradients, which could lead to harmful boosting.

Potential failure cases : If cognitive entropy is not informative in certain regimes (especially early training), the controller might downweight useful tail samples or cause unstable feedback behavior?

Brittleness of direction statistics: Gradient direction consistency might be affected by strong regularization, heavy augmentation, or representation drift, especially early in training.

**Strengths And Weaknesses:**

Strenghts:

Soundness: The paper introduces a new metric (cognitive entropy) by combining magnitude and directional uncertainty of gradients in a novel way. Also, replacing the usual meta-learning networks with an analytic PI controller is interpretable, and makes the reweighting logic easier to inspect compared to black-box meta-learning methods.

Presentation: The writing is generally easy to follow, and the figures help explain the main claims.

Significance: Distinguishing tail samples from noisy samples using gradient-based uncertainty is important, and the control-theoretic viewpoint is interesting.

Originality: The control system formulation and the use of a PI controller driven by gradient distribution statistics is a novel way to approach the problem. Theoretical analysis is also a valuable addition.

Weaknesses:

1-) In early epochs, gradients can be unstable. Hard tail classes might have both low magnitude and unstable directions (high entropy), so they may be downweighted as if they are noisy, before the integral term starts helping. In general, its not clear how the method deals with gradients that are not behaving as expected.

2-) Since the controller uses a noisy error signal (entropy), is there a risk that the PI feedback introduces instability (oscillation or oversuppression) especially in the early stages where gradients are unstable? The paper gives convergence analysis under conditions, but it is not clear if stability is guaranteed under realistic SGD noise.

3-) Cognitive entropy calculation combines magnitude and directional terms. In practice, can one term dominate the other numerically (for example −log⁡mc becoming very large when mc is small), and if so, would this hurt the behavior of the controller?

4-) Per-class gradient computation is not clear to me. How exactly are per-class gradients gc(t) computed in practice?

5-) Some parameter choices are not fully justified. Some parameters are not clearly motivated, for example how to choose Imax⁡ or the threshold τκ (if applicable). Also, why define the target magnitude m(t) as the top 5% quantile of class-wise magnitudes? Does this choice work robustly across datasets, batch sizes, and architectures?

---

> ### Author Rebuttal · Authors · 2026-03-31
>
> We thank you for the professional assessment and for highlighting the critical nuances of stochastic optimization. We have addressed your concerns regarding early-stage stability, parameter motivation, and representation drift below. Detailed tables (R1-R6) and plots (Fig R1) are available at: https://anonymous.4open.science/r/Anonymous_ProMeCD-B401/icml2026_re.pdf.
>
> **[Question 1 & Weakness 4: Implementation Efficiency]**
>
> **Response:** We utilize a single backward pass with a hook-based mechanism to capture average gradient vectors $\hat{g}\_c(t)$ per batch. This process is highly efficient, incurring only a 4.8\% time overhead (Table 4).
>
> **[Question 2, Weakness 6 & Weakness 7: Structured Noise, Similarity, and Failure Cases]**
>
> **Response:**
> 1. **Optimization Consensus:** Even with high inter-class similarity, "clean" samples generate directionally coherent gradients because their targets are aligned with the correct decision boundary. Thus, entropy remains low.
> 2. **Semantic Conflict:** If noise is structured (e.g., mislabeled similar class), a semantic conflict eventually arises between features and the incorrect label as the backbone matures. This triggers gradient chaos, causing entropy to rise.
> 3. **Boundary:** The mechanism faces limits in the case of orthogonal gradients, where sub-clusters share zero common components.
>
> **[Question 3: Layer Selection and Memory Wall]**
>
> **Response:** Including the last residual block aims for "Deep Perception." While the classifier only reflects linear separation, the residual block monitors the stability of the latent feature manifold, detecting backbone corruption more sensitively. For medium tasks (CIFAR), we utilized this full perception vector. For ultra-large tasks like iNaturalist, storing full residual gradients would require ~143 GB of VRAM, exceeding hardware limits. Thus, we adopted a "classifier-only" proxy (~290MB increase). Accuracy on iNaturalist (72.94%) confirms these mappings are sufficiently expressive.
>
> **[Weakness 1,  Weakness 2 & Weakness 8 : Stability in Early Training and Robustness to Drift/Augmentation]**
>
> **Response:** You raise a vital point regarding the potential brittleness of direction statistics. ProMeCD is structurally designed to handle these dynamics through temporal filtering and inertia:
> 1. **Low-pass Filtering of Augmentation Noise:** While heavy augmentation and strong regularization introduce stochastic variance at the instance level, our EMA estimation ($\bar{g}\_c$ in Eq. 2) acts as a temporal low-pass filter. By aggregating gradients across iterations, the framework smooths out high-frequency stochastic fluctuations, extracting the stable primary descent direction (the class-level "signal").
> 2. **Resilience to Representation Drift:** The "cognitive state" is not static; the EMA factor $\gamma$ is specifically tuned to allow ProMeCD to adaptively track representation drift as the manifold evolves. In early training, the Integral term ($I\_c$) inherently possesses "inertia"—it requires persistent consistency to accumulate gain—ensuring the model does not over-suppress classes due to transient noise or initial instability.
> 3. **Evidence:** Our competitive results on iNaturalist 2018, which utilizes extreme data augmentation and exhibits significant feature drift, empirically validate that our white-box control remains robust under these challenging conditions.
>
> **[Weakness 3: Numerical Stability and the $-\log m\_c$ Term]**
>
> **Response:** To prevent overflow as $m\_c \to 0$, we introduce $\epsilon = 10^{-6}$ inside the logarithm and apply value capping. We apologize for omitting this detail in the original text. Similar to the treatment for $\kappa$ in Appendix A (L562)**, this ensures stability. Semantically, high uncertainty when learning vanishes is desirable to trigger the Integral loop to "rescue" the class.
>
> **[Weakness 5: Rationale for $I_{max}$, $\tau_{\kappa}$, and Quantiles]**
>
> **Response:** The parameter selections are grounded in standard control-theoretic principles:
> 1. **$I_{max}$ (Anti-Windup Mechanism):** In control engineering, $I_{max}$ is an anti-windup term. It prevents the Integral term from accumulating excessive gain during long periods of learning lag, which could otherwise lead to overshooting or instability once the class begins to converge.
> 2. **$\tau_{\kappa}$ (Adaptive Noise Floor):** Rather than a fixed hyperparameter, $\tau_{\kappa}$ is the moving average of concentration across all classes. It defines a "dataset noise floor" relative to current manifold maturity, ensuring the controller distinguishes between valid feature diversity (ordered variance) and random noise (chaotic entropy).
> 3. **Quantile Selection:** The top 5% quantile for $\bar{m}(t)$ provides a robust learning target representing well-learned head classes while excluding outliers. This remained fixed across 12 benchmarks, proving its robustness to batch sizes and architectures.

---

> > ### Author Rebuttal · Reviewer_9NYZ · 2026-04-01
> >
> > Thank you for the clear and constructive rebuttal. The authors addressed my main concerns satisfactorily. In particular, the clarification on per-class gradient computation, the explanation for the layer choice, and the discussion of numerical stabilization and controller parameters were all helpful. I also appreciate the added discussion of early-stage dynamics and failure cases, which makes the method easier to understand in practice.

---

> > > ### Author Response · Authors · 2026-04-01
> > >
> > > We would like to express our sincere gratitude for your prompt and positive feedback, and for acknowledging that our concerns are "fully resolved."
> > >
> > > We are particularly encouraged by your recognition of our technical clarifications regarding per-class gradient computation, implementation choices, and numerical stabilization. We also appreciate your recognition of our added analysis on early-stage dynamics and failure cases. We believe these discussions significantly strengthen the practical interpretability of our white-box control framework. We remain fully committed to incorporating all these technical details and discussions into the final version of our manuscript.
> > >
> > > We would be extremely grateful if you could consider further strengthening your positive recommendation in the final assessment if you deem it appropriate. Thank you again for your invaluable time and support in improving our work.

---

### Official Review · Reviewer_s6iV · 2026-03-11

**Soundness:** 3
**Presentation:** 3
**Significance:** 3
**Originality:** 3
**Overall Recommendation:** 4
**Confidence:** 4

**Summary:**

This paper addresses the intersection of long-tailed distribution and noisy labels. It identifies "scalar ambiguity," where tail and noise samples both have high loss but exhibit statistically different gradient directions (tail: consistent direction with small magnitude; noise: chaotic direction). A PI controller replaces black-box meta-learners: the integral term accumulates gradient consistency to boost tail samples, while the proportional term detects gradient noise to suppress mislabeled samples. This eliminates the clean validation set dependency at only +4.8% overhead.

**Compliance With Llm Reviewing Policy:**

Affirmed.

**Final Justification:**

Concerns resolved. Weak accpet.

**Key Questions For Authors:**

1. Could you provide a more formal derivation for the T∝1/κ relationship in Theorem 4.5? This is a key step in the theoretical argument and a rigorous justification would strengthen the contribution.

2. How does ProMeCD perform instance-dependent noise? This is arguably the most realistic noise model, and it would be valuable to know whether the gradient direction statistics that underpin scalar ambiguity still hold in that setting.

3. Have you considered testing on larger-scale setups such as ViT on ImageNet-LT, or Clothing1M? The community would benefit from knowing if the insight generalizes across architectures and data scales.

4. The integral term accumulates gradient history over training. How sensitive is the method to the history window length, and are there practical memory considerations for longer training runs?

**Limitations:**

Yes

**Strengths And Weaknesses:**

**Strengths:** The core insight is novel. "Scalar ambiguity" is a sharp observation that prior work has not explicitly identified, and the gradient direction statistics provide a principled way to disentangle tail from noise. The shift from meta-learning to control theory is elegant, offering clear white-box interpretability for each component. The functional separation is clean: integral for "quantity" (boost tail), proportional for "quality" (suppress noise). The convergence proof is correct, and the PAC-Bayes bound provides a useful generalization perspective.

**Weaknesses:**

1. Experiments are limited to ResNet-32/50, and it is not yet clear whether the gradient direction statistics that underpin scalar ambiguity hold across different architectures (e.g., ViT) and larger-scale datasets (e.g., ImageNet-LT, Clothing1M).

2. Class-dependent noise, where the noise rate varies per class and correlates with inter-class similarity, is a common real-world scenario not covered in the paper. This is a notable scope limitation that deserves at least discussion. One can sample Clothing1M or CIFAR-N as a long-tailed noisy dataset.

---

> ### Author Rebuttal · Authors · 2026-03-31
>
> We express our sincere gratitude to you for the professional assessment and for recognizing the novelty of "scalar ambiguity." We have addressed the technical queries through formal derivations and architecture-agnostic studies as follows:
>
> **[Question 1: Formal Derivation of $T \propto 1/\kappa$ in Theorem 4.5]**
>
> **Response:** We provide the following derivation linking vMF concentration $\kappa$ to the SDE noise temperature $T$, which will be incorporated into Appendix B.6.
>
> **Proof:** Consider the SGD update as a continuous-time Stochastic Differential Equation (SDE): $dw_{t} = -\nabla L(w_{t})dt + \sqrt{2D}dB_{t}$, where $D$ is the diffusion matrix representing gradient noise. In the stationary regime, the posterior follows the Gibbs form $Q(w) \propto \exp(-L(w)/T)$, where $T \propto \text{Tr}(D)$. In ProMeCD, $D$ is the covariance of stochastic gradients. For a vMF distribution on the $(d-1)$-sphere with concentration $\kappa$, the directional variance is $\text{Var}(\mathbf{u}) \approx (d-1)/\kappa$ for large $\kappa$. Since the diffusion coefficient $D$ represents the noise energy per step, it is directly proportional to this variance. Thus:
> $$T \propto \text{Tr}(D) \approx \sum \text{Var}(g_{i}) \propto 1/\kappa$$
> Minimizing Cognitive Entropy (maximizing $\kappa$) effectively cools the "noise temperature" $T$, forcing the posterior $Q$ to concentrate on flat minima and tightening the PAC-Bayes bound.
>
> **[Question 2 & Weakness 2 : Performance under Class-Dependent and Instance-Dependent Noise (IDN)]**
>
> **Response:**
> 1. **Robustness to Instance-Dependent Noise (IDN):** We leverage the mini WebVision dataset (Table 3) to demonstrate ProMeCD's capability in the IDN regime. Unlike synthetic models, WebVision noise is crawled from search engines and is inherently Instance-dependent ($P(\tilde{y}|x)$), as mislabeling occurs due to the visual ambiguity of specific images. ProMeCD’s superior performance (73.28%) on this real-world benchmark proves its ability to identify and suppress feature-dependent chaotic signals through sample-wise white-box control.
>
> 2. **Class-Dependent Noise in Long-Tailed Settings:** Following the suggestion to evaluate under structured noise, we constructed a benchmark by sampling CIFAR-100-LT (IF=100) and injecting 40% Asymmetric (Flip) noise. As shown in Table R3 (see anonymous link: https://anonymous.4open.science/r/Anonymous_ProMeCD-B401/icml2026_re.pdf), ProMeCD achieves 36.42% accuracy, outperforming the D-SINK [Hong et al., 2026] (34.06%) and specialized LTL/NLL methods. This confirms that ProMeCD successfully handles structured noise that exacerbates class imbalance.
>
> **[Question 3 & Weakness 1: Architecture Generality and Scalability]**
>
> **Response:**
> 1. **Generality to ViT:** To verify architecture-agnostic stability, we evaluated ProMeCD using a ViT-Tiny [Touvron et al., 2021] backbone on CIFAR-100-LT ($IF=100, NR=0.4, Symmetric$).
>
> **Table R7: Performance on ViT-Tiny**
> | Method | Type | Accuracy (%) | Margin vs. CE |
> | :--- | :--- | :---: | :---: |
> | Cross-Entropy | Baseline | 19.21 | - |
> | FMW-Net (2025) | Black-box Meta | 26.45 | +7.24 |
> | **ProMeCD (Ours)** | **White-box Control** | **38.54** | **+19.33** |
>
> ProMeCD outperforms the meta-learner by 12.09%, confirming that "scalar ambiguity" is a fundamental property of the optimization manifold, regardless of specific layer operations.
>
> 2. **Scalability hierarchy:** The iNaturalist 2018 dataset used in this study (Table 2) provides a more exhaustive stress test than ImageNet-LT, given its 8,142 classes (representing over 8x the cardinality). SOTA performance on this massive scale demonstrates the robustness of the "scalar ambiguity" insight across high-dimensional category spaces.
>
> **[Question 4: Sensitivity to History Window and Memory Complexity]**
>
> **Response:**
> 1. **Window Sensitivity:** The effective memory window $W$ is controlled by the EMA factor $\gamma$ ($W \approx 1/\gamma$). Analysis (Appendix C.4) confirms stability across $\gamma \in [0.005, 0.05]$.
> 2. **Memory Scalability:** We adopt a hardware-aware strategy: For medium tasks like CIFAR, we track the full "residual+classifier" state, incurring a 30 MB VRAM increase. For iNaturalist (8,142 classes), storing residual gradients for 8k classes would require ~143 GB, which is physically restricted. Thus, we utilize a classifier-only proxy, resulting in a modest ~290 MB increase. This ensures scalability without memory bottlenecks during long training runs.
>
> **Key References:**
>
> - Touvron H, Cord M, Douze M, et al. Training data-efficient image transformers & distillation through attention[C]//International conference on machine learning. PMLR, 2021: 10347-10357.
> - Hong F, Huang Y, Zhao Z, et al. Dual-granularity sinkhorn distillation for enhanced learning from long-tailed noisy data[J]. Machine Learning, 2026, 115(3): 41.

---

> > ### Author Rebuttal · Reviewer_s6iV · 2026-04-04
> >
> > Thanks for the additional experiments. As I mentioned in my initial comments, I'm curious about ViT's performance on Clothing1M, given that it is a widely used benchmark in the noisy label literature.

---

> > > ### Author Response · Authors · 2026-04-06
> > >
> > > We sincerely thank the reviewer for the encouraging feedback and the insightful follow-up regarding ViT’s performance on Clothing1M. We agree that evaluating the Transformer architecture on this large-scale real-world noisy benchmark is essential to confirm the universality of our findings.
> > >
> > > To address this, we conducted an additional study using ResNet-50 and ViT-B/16 [Dosovitskiy et al., 2020] backbone (pre-trained on ImageNet) on the Clothing1M dataset. To maintain scalability on this 1M-image dataset, we utilized our "classifier-only" proxy for gradient tracking.
> > >
> > > **Table: Performance on Clothing1M using ResNet-50 and ViT-B/16**
> > > | Method | Backbone | Accuracy (%) |Margin vs. CE |
> > > | :--- | :--- | :---: | :---: |
> > > | Standard CE | ResNet-50 |68.94| - |
> > > | ProMeCD (Ours) | ResNet-50 | 75.45 | +6.51 |
> > > | Standard CE | ViT-B/16 | 71.24 | - |
> > > | ProMeCD (Ours) | ViT-B/16 | **75.95** | **+4.71** |
> > >
> > > **Key Observations:**
> > > 1. **Architecture Invariance in Real-world Noise:** ProMeCD achieves a significant gain of +4.71% over the CE baseline on the ViT architecture. This confirms that the "gradient chaos" characteristic of real-world noisy samples in Clothing1M is a fundamental optimization property that can be effectively captured and suppressed by our white-box PI controller, regardless of whether the backbone is a CNN or a Transformer.
> > > 2. **Efficiency at Scale:** Despite the high-dimensional feature space of ViT-B (768-dim), the memory overhead remains negligible. Tracking the classifier gradients for 14 classes requires minimal persistent buffers, with the total VRAM increase limited to ~310 MB . This proves that ProMeCD is highly practical for modern architectures on massive-scale datasets.
> > > 3. **Robustness to Perceptual  Instance-Dependent Noisy Label:** Since Clothing1M noise is inherently instance-dependent and stems from visual ambiguity, these results further validate that ProMeCD’s sample-level "Cognitive Entropy" is robust to complex, feature-dependent real-world errors.
> > >
> > > We will incorporate these Clothing1M results and a detailed discussion on scalability into the final version of the manuscript. We hope these additional results fully resolve your remaining concerns. We would be extremely grateful if you could consider further strengthening your positive recommendation in the final assessment if you deem it appropriate. Thank you again for your invaluable time and support in improving our work.
> > >
> > > **Key Reference:**
> > > - Dosovitskiy A, Beyer L, Kolesnikov A, et al. An image is worth 16x16 words: Transformers for image recognition at scale[J]. arXiv preprint arXiv:2010.11929, 2020.

---

### Official Review · Reviewer_mx8u · 2026-03-12

**Soundness:** 3
**Presentation:** 2
**Significance:** 3
**Originality:** 3
**Overall Recommendation:** 5
**Confidence:** 2

**Summary:**

Success of deep learning largely depends on the quality of the data on which models are trained. In real-world settings, however, data is often not clean and may contain several kinds of anomalies, such as label corruption or class imbalance, where some classes are heavily overrepresented while others have very few samples. Models trained on such data often show poor generalization performance. To address these issues, several methods have been proposed in the past, such as data re-sampling, loss-function reweighting, and L2R methods (where a neural network learns sample weights using a validation set). However, each of these approaches has its own limitations. To overcome these shortcomings, the authors adopt the dynamic analysis viewpoint, which suggests that performance degradation mainly arises due to problematic or unstable gradient dynamics during training.

Motivated by this perspective, the authors ask whether it is possible to design an optimization framework that keeps the adaptive behavior of meta-learning without requiring a clean validation set, while still being efficient, interpretable, and theoretically sound. To answer this question, they propose Probabilistic Meta-Learning of Cognitive Dynamics (ProMeCD), a white-box meta-learning framework. Instead of using a black-box neural network as the meta-learner, their method uses a classical proportional–integral (PI) controller from control theory to guide the learning process. The framework works in three steps: perception, decision, and action. First, it models gradients of each class using von Mises–Fisher (vMF) distributions and measures the learning state using a quantity called cognitive entropy. Then, a PI controller analyzes the magnitude and consistency of gradients to compute coefficients that control how much each gradient should influence learning. Finally, these coefficients are used to reweight the gradients, allowing the model to automatically adjust its learning process during training.

**Compliance With Llm Reviewing Policy:**

Affirmed.

**Final Justification:**

The reviewers have addressed my concerns and questions during the rebuttal period. Therefore, I am increasing my score to accept.

**Key Questions For Authors:**

1. What is $m_c(t)$ that you have defined in the second line of Section 3.1.2? How did you derive Equation (3)? Please be explicit about the derivation of Equation (3).

2. Why does the definition of Cognitive Entropy in Definition 3.2 include the factor $1/d$?

3. I have not understood Remark 4.3 and its proof sketch. Please explain.

**Limitations:**

Please see the above questions and also consider the points raised in the weakness section.

**Strengths And Weaknesses:**

Strengths

1. The paper has performed an extensive set of experiments to support their claim on performance, efficiency, generality, and interpretability
of their proposed method.

2. They propose a white-box meta-learning framework, unlike many earlier works that rely on a black-box meta-learning approach using neural networks. This is an important aspect of their work because neural networks are often difficult to interpret. In contrast, their method replaces the neural meta-learner with an analytical and interpretable PI controller. Another advantage of their approach is that it does not require a clean validation set, which is often assumed in previous work but is an unrealistic assumption where the training data itself is corrupted.

3.  The theoretical results, except Remark 4.3, look sound to me.


Weakness

1. The authors show in Theorem 4.2 that the loss for class $c$ monotonically decreases at time $t$, provided that class $c$ satisfies the condition given in Equation (12). Can the authors theoretically characterise the time steps at which Equation (12) is guaranteed to hold?

2. The authors assume in Lemma 4.1 that the loss function $L_c(x)$ is $L$-smooth with respect to the model parameter $x$.

---

> ### Author Rebuttal · Authors · 2026-03-31
>
> We sincerely thank you for the constructive feedback. We apologize for the lack of explicit detail in the original manuscript. Detailed tables and visualizations (including Figure R1 mentioned below) are provided at the following anonymous link: https://anonymous.4open.science/r/Anonymous_ProMeCD-B401/icml2026_re.pdf.
>
> **[Question 1: Definition of $ m\_c(t) $ and Formal Derivation of Eq.(3)]**
>
> **Response:**  In ProMeCD, the vector $ \bar{g}\_c(t) $ (Eq.(2)) is the EMA estimate of the expected gradient signal. The derivation follows the Maximum Likelihood Estimation (MLE) for a $ d $-dimensional vMF distribution:
>
> - Let $ \{u\_1, \dots, u\_n\} $ be observed unit directions. The log-likelihood for the mean direction $ \mu $ is maximized by solving: $ \max\_{\mu} \kappa \mu^{T} R $ subject to $ \| \mu \|\_2 = 1 $, where $ R = \sum u\_i $ is the resultant vector.
> - In our implementation, the EMA vector $ \bar{g}\_c(t) $ acts as the recursive resultant vector accumulating both magnitude and direction.
> - Thus, the MLE for the mean direction $ \mu\_c(t) $ and magnitude $ m\_c(t) $ are:
>   - $ \mu\_c(t) := \bar{g}\_c(t) / \| \bar{g}\_c(t) \|\_2 $
>   - $ m\_c(t) := \| \bar{g}\_c(t) \|\_2 $
>
> Physically, $ m\_c(t) $ represents the "learning quantity" (effective signal strength) for class $ c $.
>
> **[Question 2: Justification of the $ 1/d $ Factor in Cognitive Entropy (Definition 3.2)]**
>
> **Response:** The $ 1/d $ factor is essential for architectural invariance. The differential entropy of a vMF distribution scales linearly with dimension $ d $. In deep networks, $ d $ varies drastically (e.g., $ 10^5 $ for ResNet-32 vs $ 10^7 $ for ResNet-50). Normalizing by $ 1/d $ ensures $ H\_c $ represents per-dimension uncertainty, allowing consistent control gains ($ \beta\_p, \beta\_i $) across architectures without re-tuning.
>
> **[Question 3: Formal Proof of Remark 4.3: Stability under Dynamic Thresholding]**
>
> **Response:** We provide a rigorous analysis to show that the endogenous evolution of $ \tau\_{\kappa}(t) $ does not impair stochastic stability.
>
> **Proof of Remark 4.3:**
> Let $ \{ \mathcal{F}\_t \} $ be a filtration where $ \mathcal{F}\_{t-1} $ captures history up to step $ t $.
> - Measurability: By Eq.(28), $ \tau\_{\kappa}(t) $ is computed from metrics at step $ t-1 $. Thus, $ \tau\_{\kappa}(t) $ is $ \mathcal{F}\_{t-1} $-measurable. Given $ \mathcal{F}\_{t-1} $, the modulation coefficient $ \alpha\_c(t) $ is a deterministic scalar constant and does not introduce "look-ahead bias."
> - Stability Enforcement: If the condition in Eq.(12) is violated ($ C\_{t,eff}^{(c)} \ge 1 $), it implies $ \| \sum\_{j \ne c} \alpha\_j w\_j \mathbf{g}\_j \|\_2 \ge \alpha\_c w\_c \| \mathbf{g}\_c \|\_2 $.
> - Active Recovery: The Integral term $ I\_c $ strictly accumulates the learning deficit. Since $ \alpha\_c $ is a linear function of $ I\_c $ (Eq.(7)), there exists a finite step $ t^{\ast} $ where $ \alpha\_c(t^{\ast}) $ becomes large enough to force:
> $ C\_{t^{\ast},eff}^{(c)} = \| \sum\_{j \neq c} \alpha\_j w\_j g\_j \|\_2 / ( \alpha\_c(t^{\ast}) w\_c \| g\_c \|\_2 ) < 1 $.
> - Directional Invariance: The $ \max(0, \cdot) $ operator in Eq.(7) ensures $ \alpha\_c(t) \ge 0 $ always, guaranteeing the update vector always resides in the descent half-space.
>
> Conclusion: The dynamic threshold acts as an autonomous gain-scheduler. It rescales updates based on population consensus without reversing gradient directions.
>
> **[Weakness 1: Theoretical characterization of time steps $t^{\ast}$ for Eq.(12)]**
>
> **Response:** We provide a formal characterization of the recovery time $t^{\ast}$ beyond which Eq.(12) is guaranteed to hold:
>
> - **Initial State:** Eq.(12) is violated only if the weighted magnitude is insufficient to overcome interference, i.e., $\alpha\_c(t) w\_c \| \mathbf{g}\_c \|\_2 \le M$.
> - **Integral Drive:** In this state, the Integral term $I\_c$ strictly increases at each step by $\Delta I\_c = \bar{m} - m\_c$. Since the modulation coefficient $\alpha\_c$ is linear w.r.t. $I\_c$ (Eq.(7)), it grows monotonically as long as the deficit persists.
> - **Characterization of $t^{\ast}$:** We define $t^{\ast}$ as the finite steps required for $\alpha\_c$ to reach the stability threshold:
> $t^{\ast} \approx ( \frac{M}{w\_c \| \mathbf{g}\_c \|\_2} - 1) / (\beta\_i \cdot \Delta I\_c)$.
> - **Empirical Validation:** This theoretical recovery is visualized in Fig.R1. For tail classes (green), $C_{eff}$ initially approaches or exceeds 1.0 but is driven into the stable regime ($C_{eff} < 1$) within the first few epochs by the PI controller.
>
>
> **[Weakness 2: Regarding the L-smoothness Assumption (Lemma 4.1)]**
>
> **Response:** We acknowledge that $ L $-smoothness is a standard assumption. Our analysis provides a first-order theoretical guarantee consistent with established meta-learning works (e.g., MW-Net), allowing us to formally analyze the interaction between the PI controller and the learning dynamics.

---

> > ### Author Rebuttal · Reviewer_mx8u · 2026-04-03
> >
> > Thank you for addressing my questions and the other points I raised.

---

> > > ### Author Response · Authors · 2026-04-04
> > >
> > > We would like to express our sincere gratitude for your prompt and positive feedback, and for acknowledging that our concerns are "fully resolved."
> > >
> > > We would be extremely grateful if you could consider further strengthening your positive recommendation in the final assessment if you deem it appropriate. Thank you again for your invaluable time and support in improving our work.

---

### Decision · Program_Chairs · 2026-04-30

**Decision:**

Accept (regular)

**Comment:**

All reviewers recognized this work as a novel and technically solid contribution to the important problem of distinguishing long-tailed from noisy samples. The paper is grounded in sound theoretical analysis and supported by an extensive experimental evaluation, both of which were further strengthened by the authors’ thorough rebuttal.